



# Atmospheric CO₂ exchanges measured by Eddy Covariance over a temperate salt marsh and influence of environmental controlling factors

Jérémy Mayen[1,2*], Pierre Polsenaere[1], Éric Lamaud[3], Marie Arnaud[1,4], Pierre Kostyrka[1,5], Jean-Marc Bonnefond[3], Philippe Geairon[1], Julien Gernigon[6], Romain Chassagne[7], Thomas Lacoue-Labarthe[8], Aurore Regaudie de Gioux[5], Philippe Souchu[2]

[1]IFREMER, Littoral, Laboratoire Environnement Ressources des Pertuis Charentais (LER/PC), BP 133, 17390, La Tremblade, France
[2]IFREMER, Littoral, Laboratoire Environnement Ressources Morbihan-Pays de Loire (LER/MPL), BP 21105, 44311, Nantes, France
[3]INRAE, Bordeaux Sciences Agro, ISPA, F-33140 Villenave d'Ornon, France
[4]Institute of Ecology and Environmental Sciences Paris (iEES-Paris), Sorbonne University, Paris 75005, France
[5]IFREMER, Dyneco, Pelagos, ZI de la Pointe du Diable - CS 10070 - 29280 Plouzané, France
[6]LPO, Réserve Naturelle de Lilleau des Niges, 17880, Les Portes en Ré, France
[7]BRGM, 3 avenue Claude-Guillemin, BP 36009, 45060 Orléans, Cedex 02, Orléans, France
[8]Littoral Environnement et Sociétés (LIENSs), UMR 7276, CNRS, La Rochelle Université, 2 Rue Olympe de Gouge, 17000 La Rochelle, France

*Correspondence to*: Jérémy Mayen (jeremy.mayen@ifremer.fr)

**Summary.** We deployed an atmospheric eddy covariance system to measured continuously the net ecosystem CO₂ exchanges (NEE) over a salt marsh and determine the major biophysical drivers. Our results showed an annual carbon sink mainly due to photosynthesis of the marsh plants. Our study also provides relevant information on NEE fluxes during marsh immersion by decreasing daytime CO₂ uptake and night-time CO₂ emissions at the daily scale whereas the immersion did not affect the annual marsh C balance.



**Abstract.** Within the coastal zone, salt marshes are atmospheric $CO_2$ sinks and represent an essential component of biological carbon (C) stored on Earth due to a strong primary production. Significant amounts of C are processed within these tidal systems which requires a better understanding of the temporal $CO_2$ flux dynamics, the metabolic processes involved and the controlling factors. Within a temperate salt marsh (French Atlantic coast), continuous $CO_2$ exchange measurements were performed by the atmospheric eddy covariance technique to assess the net ecosystem exchange (NEE) at diurnal, tidal and seasonal scales and the associated relevant biophysical drivers. During emersion, NEE fluxes were partitioned into net ecosystem production (NEP), gross primary production (GPP) and ecosystem respiration ($R_{eco}$) to study marsh metabolic processes. Over the year 2020, the measured net C balance was -483 g C $m^{-2}$ $yr^{-1}$ while GPP and $R_{eco}$ absorbed and emitted 1019 and 533 g C $m^{-2}$ $yr^{-1}$, respectively. The highest $CO_2$ uptake was recorded in spring during the growing season for halophyte plants in relationships with favourable environmental conditions for photosynthesis whereas in summer, higher temperatures and lower humidity rates increased ecosystem respiration. At the diurnal scale, the salt marsh was a $CO_2$ sink during daytime, mainly driven by light, and a $CO_2$ source during night-time, mainly driven by temperature, irrespective of emersion or immersion periods. However, daytime immersion strongly affected NEE at the daily scale by reducing marsh $CO_2$ uptake up to 90%. During night-time immersion, $CO_2$ emissions could be completely suppressed, even causing a change in metabolic status from source to sink under certain situations, especially in winter when $R_{eco}$ rates were lowest. At the annual scale, tidal rhythm did not significantly affect the net C balance of the studied salt marsh since similar annual values of measured NEE and estimated NEP were recorded.

## 1. Introduction

Salt marshes are intertidal coastal ecosystems dominated by salt-tolerant herbaceous plants located at the terrestrial-aquatic interface. Despite their low surface area at the global scale (54650 $km^2$; Mcowen et al., 2017), salt marshes provide important ecosystem services such as erosion protection (natural buffer zones), a water purification, a nursery for fisheries (Gu et al., 2018) and a high capacity for atmospheric $CO_2$ uptake and carbon (C) sequestration in their organic matter (OM) enriched sediments and soils (Mcleod et al., 2011; Alongi, 2020). Over salt marshes, emersion at low tide and slow immersion at high tide favour this $CO_2$ fixation through photosynthesis of terrestrial and aquatic vegetations and also a strong benthic-pelagic coupling (Cai, 2011; Wang et al., 2016; Najjar et al., 2018). The high net primary production (NPP) rate of salt marshes on the Atlantic Coast of the United States (1070 g C $m^{-2}$ $yr^{-1}$; Wang et al., 2016) makes marshes one of the most productive ecosystems on Earth (Duarte et al., 2005; Gedan et al., 2009). According to Artigas et al. (2015), approximately 22% of C fixed through this marsh NPP is then buried in coastal sediments as "blue C" thus allowing salt marshes to be a large biological C pool (Chmura et al., 2003; Mcleod et al., 2011). However, tidal immersion can generate strong lateral exports of organic and inorganic C to the coastal ocean (Wang et al., 2016), partly favouring atmospheric $CO_2$ emissions from adjacent coastal ecosystems downstream (Wang and Cai, 2004; Jiang et al., 2008). Salt marshes represent an





biogeochemically active interface area within the coastal zone but are also threatened by sea level rise, erosion and global warming (Gu et al., 2018) which could significantly alter their capacity to sink and store C (Campbell et al., 2022). Thus, atmospheric $CO_2$ exchanges need to be accurately measured and better understood, especially the influence of biotic and

abiotic controlling factors, in order to be included in regional and global C budgets (Borges et al., 2005; Cai, 2011) and to predict future mash C sinks within the context of climate change.

In temperate salt marshes, actual and historical land and water management, plant species, tidal influence and environmental conditions have shown to play an important role in their C cycle. Generally, strong seasonal variations in the net ecosystem $CO_2$ exchange (NEE) were recorded with a marsh $CO_2$ sink during the hottest and brightest months and a $CO_2$

source during the rest of the year (Schäfer et al., 2014; Artigas et al., 2015). At a smaller scale, in urban salt marshes (USA), the highest $CO_2$ uptake occurred at midday in general whereas the system emitted $CO_2$ throughout the night-time, illustrating the major role of net radiations in the marsh metabolic status (Schäfer et al., 2014, 2019). Tidal immersion over salt marshes can also strongly influence both daytime and night-time NEE fluxes, especially during spring tides (Forbrich and Giblin, 2015). For instance, negative correlations between NEE and tidal effects were computed in a temperate salt marsh (USA)

with *Spartina alterniflora* and *Phragmites australis*, especially in summer and winter, with negative (sink) and positive (source) NEE values during incoming and ebbing tides, respectively (Schäfer et al., 2014). Wang et al. (2006) showed a competitive advantage for the growth and productivity of *S. alterniflora* plants under a moderate level of salinity (i.e. 15‰) and immersion conditions. These different EC studies highlight the complexity of the C cycle over salt marshes and the associated biophysical factors driving $CO_2$ fluxes that require more *in situ* and integrative NEE measurements within and

between all compartments at the different temporal scales to better understand the biogeochemical functioning of these ecosystems under changing sea-level conditions.

Within coastal wetlands such as salt marshes and tidal bays, $CO_2$ fluxes at sediment-air interfaces can be accurately assessed with static chambers by repeating measurements over different intertidal habitats (Xi et al., 2019; Wei et al., 2020a). Yet, a major limitation of this method is that it can hardly include the temporal and spatial $CO_2$ flux variability

across different vegetations and habitats (Migné et al., 2004). Atmospheric $CO_2$ fluxes can also be performed in heterogeneous tidal systems using the atmospheric eddy covariance (EC) technique based on the covariance between fluctuations in the vertically velocity and air $CO_2$ concentration (Baldocchi et al., 1988; Aubinet et al., 1999; Baldocchi, 2003). This direct and non-invasive micrometeorological technique has been of growing interest over the coastal zone to assess the NEE through accurate, continuous and high-frequency $CO_2$ flux measurements (Schäfer et al., 2014; Artigas et al.,

2015; Forbrich and Giblin, 2015). This method has been deployed over blue carbon systems such as mangroves (Rodda et al., 2016; Gnanamoorthy et al., 2020), seagrass meadows (Polsenaere et al., 2012; Van Dam et al., 2021) and salt marshes (Artigas et al., 2015; Forbrich et al., 2018; Schäfer et al., 2019) to assess their $CO_2$ uptake capacity. In intertidal systems like salt marshes, the major advantage of the EC method is to measure the NEE at the ecosystem scale, coming from all habitats



inside the footprint, at various time scales from hours to years and at both the sediment/air and water/air interfaces (i.e. low and high tides, respectively) (Kathilankal et al., 2008; Wei et al., 2020b). Although many studies have used this method to study tidal effects on NEE fluxes over salt marshes, only a limited number have looked at the loss of $CO_2$ uptake due to tidal effects. Moreover, marsh metabolic fluxes (NEE) can be partitioned through modelling approaches into net ecosystem production (NEP), gross primary production (GPP) and ecosystem respiration ($R_{eco}$) (Kowalski et al., 2003; Reichstein et al., 2005; Lasslop et al., 2010). However, use of the EC method requires significant qualitative and quantitative processing and data correction applied to each specific site since this method relies on the physical and theoretical backgrounds (Baldocchi et al., 1988; Burba, 2021) and is adapted (technically and scientifically) to the coastal systems.

Our study focused on the atmospheric $CO_2$ uptake capacity of a tidal salt marsh (old anthropogenic marsh) under the influence of biophysical factors and its potential role in global and regional C budgets. For this purpose, we deployed an atmospheric eddy covariance (EC) station to measure vertical $CO_2$ fluxes (NEE) over the year 2020 at the ecosystem scale on the Bossys perdus salt marsh on Ré Island connected to the French continental shelf of the Atlantic Ocean. Here, we aim to (a) describe NEE flux temporal series measured at different temporal scales (diurnal, tidal and seasonal scales) using the EC technique, (b) evaluate the relevant environmental factors that control atmospheric $CO_2$ exchanges (i.e. NEE) and (c) accurately qualify and quantify the effects of tides on the marsh $CO_2$ metabolism.

## 2. Materials and methods

### 2.1. Study site

The study was conducted at the Bossys perdus salt marsh situated along the French Atlantic coast on Ré Island (Fig. 1). It corresponds to a vegetated intertidal area of 52.5 ha that has been protected inside the National Natural Reserve (NNR) since 1981 (Fig. 1). Since the 17th century, the salt marsh has experienced successive periods of intensive land-use (salt harvesting, oyster farming) and returns to natural conditions before becoming part of the NNR. It is currently managed to restore its natural dynamics (Gernigon, personal communication). This salt marsh is linked to the Fier d'Ars tidal estuary that exchanges between 2.4 and 10.2 million $m^3$ of coastal waters with the Breton Sound continental shelf (Bel Hassen, 2001). This communication allows to (1) drain the intertidal zone of the estuary including mudflats (slikke) and tidal salt marshes (schorre) and (2) supply coastal water to a large complex of artificial salt marshes (i.e. salt ponds) located upstream of the dyke (Fig. 1). The artificial marsh waters preserved and managed by the NNR for biodiversity protection (Mayen et al., submitted) are eventually flushed back to the estuary downstream through the Bossys perdus marsh (Fig. 1).



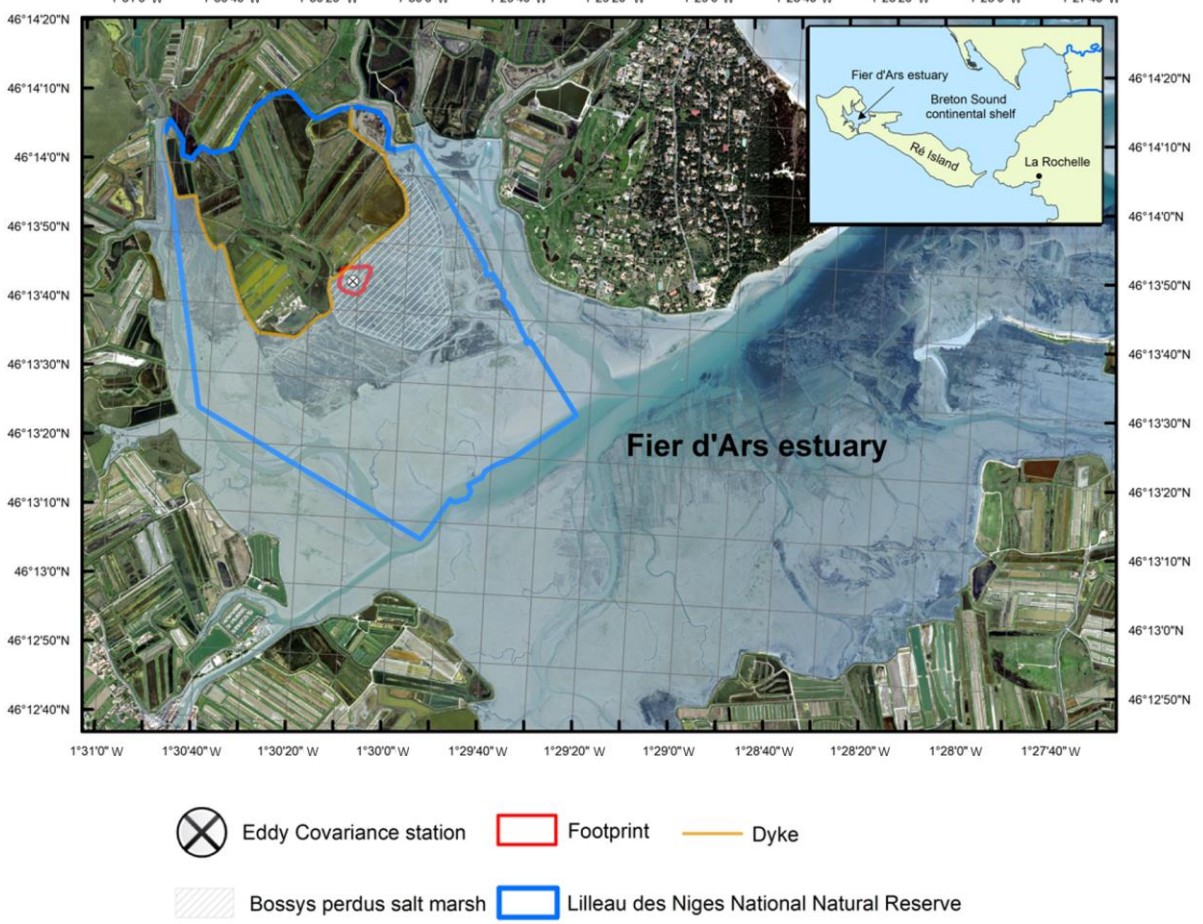

**Figure 1: The studied Bossys perdus salt marsh located on the French Atlantic coast within the National Natural Reserve (blue line**
**delimitation) on Ré Island. The salt marsh is connected to the Fier d'Ars tidal estuary (light blue). The dyke separates terretrial**
**and maritime marsh areas (orange line). The eddy covariance system and associated estimated footprint are indicated (black cross**
**and red line; see Fig. 2). From geo-referenced IGN orthogonal images (IGN 2019).**

The Bossys perdus salt marsh, located upstream of the estuary (schorre), is subjected to semi-diurnal tides from the

Breton Sound continental shelf (Fig. 1) allowing the marsh immersion by two main channels differently in space, time and

frequency according to the tidal periods (Fig. 2). At high tides and spring tides, advected coastal waters can completely fill

channels and immerge the marsh through variable water heights depending on tidal amplitudes and meteorological

conditions (Fig. A.1-C). On the contrary, during low tides, the marsh vegetation at the benthic interface is emerged into the

atmosphere without any coastal waters (Fig. A.1-A). During this time, channels allow to drain upstream artificial marsh

waters to the estuary (Fig. 2).



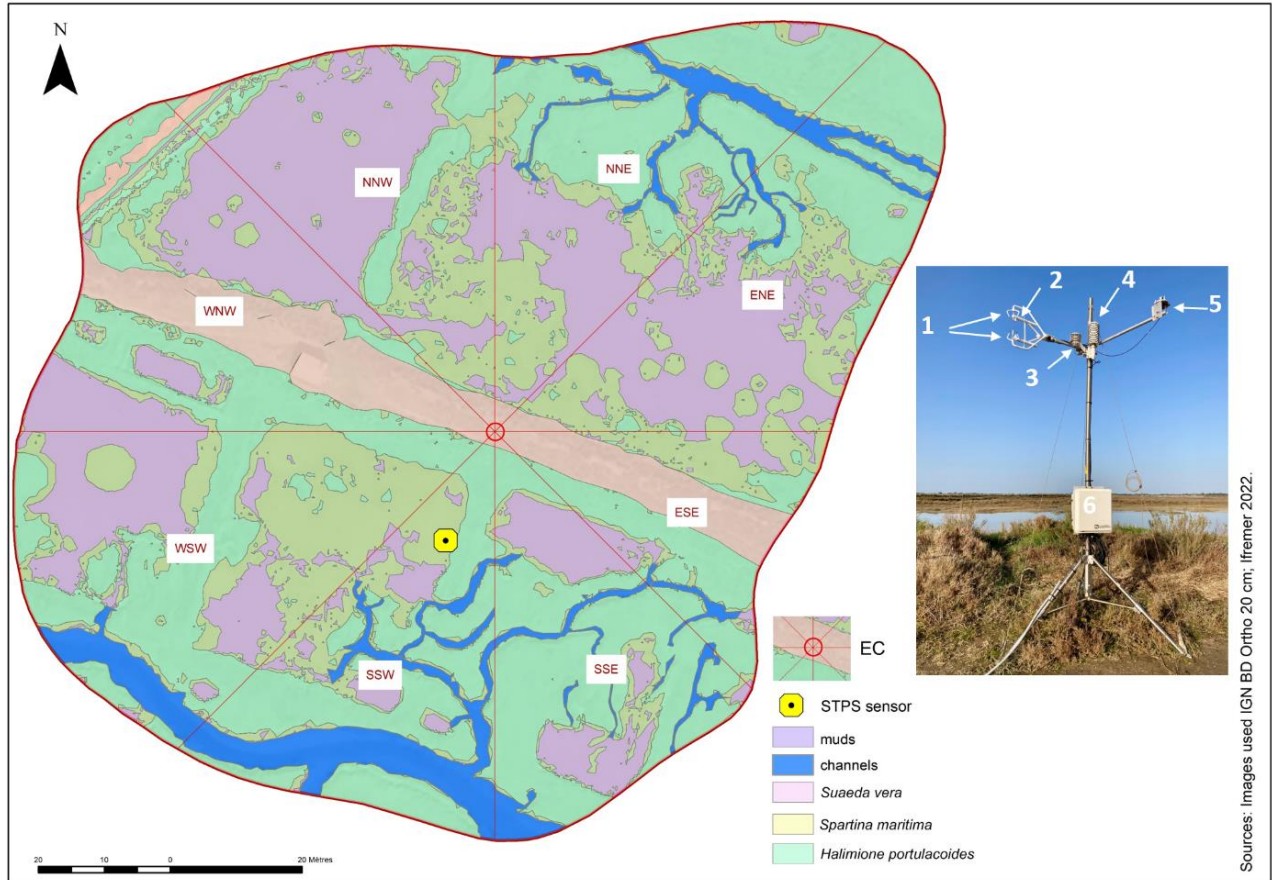

**Figure 2: Location and set-up of the eddy covariance (EC) system within the Bossys perdus salt marsh and its associated footprint estimated from Kljun et al. 2015 and averaged over the year 2020 (70% countour line, i.e. 13042 m²). Wind sectors (45°) and marsh habitats (see Table 1) are represented. The STPS sensor (in yellow), measuring water height (Hw), salinity (Sw) and temperature (Tw), was located in the SSW sector. The EC system (*Campbell Scientific*) includes (1) the ultrasonic anemometer (CSAT3), (2) the open-path infrared gas analyser (EC150), (3) the temperature probe (100K6A1A Thermistor), (4) the temperature/relative humidity sensor (HMP155A), (5) the silicon quantum sensor (SKP215), (6) the central acquisition system (CR6) and the electronics module (EC100). A rainfall sensor (TE525MM, Raingauge Texas) simultaneously measured the cumulative precipitation. From geo-referenced IGN orthogonal images (IGN 2019).**

## 2.2. Theory of the EC technique

The atmosphere is characterized by horizontal air flows that include several rotating eddies of different sizes and frequencies caused by buoyancy and shear of upward and downward moving air allowing for the transport of atmospheric gas such as $CO_2$ (Aubinet et al., 1999; Baldocchi, 2003; Burba, 2021). Each eddy has three-dimensional components including a vertical movement of the wind and each air parcel transported by eddies has its own micrometeorological characteristics (gas concentration, temperature, humidity). Thus, the eddy covariance (EC) technique can be used to measure



the vertical component of these turbulent eddies to quantify the net $CO_2$ fluxes at the ecosystem-atmosphere interface

(Burba, 2021). The averaged vertical flux of any gas ($F$, µmol m$^{-2}$ s$^{-1}$) can be expressed as the covariance between the vertical wind speed ($w$, m s$^{-1}$), air density ($\rho$, Kg m$^{-3}$) and the dry mole fraction ($s$) of the gas of interest as:

$$F = \overline{\rho\,w\,s} \approx \overline{p}\;\overline{w's'} \tag{1}$$

where the overbar represents the time average of the parameter (i.e. 10 minutes in this study due to strong fluctuations at the tidal scale) and the apostrophe indicates the instantaneous turbulent fluctuations in these parameters relative to their temporal

average (Reynolds, 1883). The Reynold's decomposition was used to break the instantaneous term down into its mean and deviation (e.g. w = $\overline{w}$ + w′) (Reynolds, 1883; Burba, 2021). Moreover, this equation (Eq. 1) is obtained by assuming, on a flat and homogeneous surface, that (1) the variation in air density is negligible, (2) there is no divergence or convergence of large-scale vertical air motion and (3) atmospheric conditions are stable and stationary (Aubinet et al., 2012). A negative flux of atmospheric $CO_2$ is directed towards the ecosystem, and is therefore characterized as a sink, and *vice versa* for positive

fluxes qualified as sources of $CO_2$ to the atmosphere.

### 2.3. Eddy covariance and micrometeorological measurements

An EC system was continuously deployed at the Bossys perdus salt marsh to measure the net $CO_2$ ecosystem exchange (NEE, µmol m$^{-2}$ s$^{-1}$). The set of EC sensors (Fig. 2), at a height of 3.15 m, was composed of an open-path infrared gas

analyser (model EC150, *Campbell Scientific Inc.*, Logan, UT) to measure the $CO_2$ (mg m$^{-3}$) and $H_2O$ (g m$^{-3}$) concentrations in the air as well as the atmospheric pressure (kPa) and an ultrasonic anemometer (model CSAT3, *Campbell Scientific Inc.*, Logan, UT) to measure the three-dimensional components of wind speed (U, V and W; m s$^{-1}$) at a frequency of 20 Hz and averaged every 10 minutes (Fig. 2). The EC150 gas analyser also measured the air temperature using a thermistor probe (model 100K6A1A Thermistor, *BetaTherm*). The EC100 electronics module (model EC100, *Campbell Scientific Inc.*)

allowed to synchronize high-frequency measurements and rapid communications between the CR6 datalogger (model CR6, *Campbell Scientific Inc.*) and EC devices including EC150 and CSAT3A (Fig. 2). The CR6 datalogger is a powerful core component for the data acquisition system. Additional meteorological data such as relative humidity (RH, %), air temperature (Ta, °C) and photosynthetically active radiation (PAR, µmol m$^{-2}$ s$^{-1}$) were recorded every 10 minutes simultaneously and at the same height as the EC sensors, by a temperature/relative humidity sensor (HMP155A, with

RAD14 natural ventilation shelter) and a silicon quantum sensor (SKP215, Skye Instruments), respectively (Fig. 2). A rainfall sensor (TE525MM, Raingauge Texas), located 10 m away and connected to the EC station, simultaneously measured the cumulative precipitation at a height of 1 m (rainfall, mm). All high-frequency EC data were recorded on a SD micro-card (2 Go, *Campbell Scientific Inc.*) that was replaced every two weeks, whereas meteorological data were recorded and stored in the central acquisition system (CR6). The EC system was connected to two rechargeable batteries (AGM, 12 volts and

260 amperes per hour) powered by a monocrystalline solar panel (Victron, 24 volts, 200Wp module with MPPT 100V/30A



controller). The EC sensors were checked and cleaned every two weeks and the EC150 was calibrated each season with a zero-air calibration of 0 ppm (*Campbell Scientific Inc.*) and a certificated $CO_2$ standard of 520 ppm (*Gasdetect*). Water height (Hw), temperature (Tw) and salinity (Sw) values were measured every 10 min., along with EC data using a STPS probe (NKE Instrumentation) located 20 m away from the EC system (Fig. 2). The sensor was checked every two months at
the laboratory to verify possible derivations in the measured parameters.

Footprints were estimated using the model of Kljun et al. (2015) applied to data from the year 2020 to obtain an annual averaged footprint from the constant measurement height (zm, 3.15 m), displacement height (d = 0.1), mean wind velocities (u_mean, m s$^{-1}$), standard deviations of the lateral velocity fluctuations after rotation [sigma_v, m s$^{-1}$], the Obukhov length (L), friction velocities (u*, m s$^{-1}$) and wind directions (°) obtained from the EC measurements and the EddyPro processing
software (EddyPro® v7.0.8, LI-COR *Inc*.) output. For all calculations (i.e. habitat coverage, relationships with $CO_2$ fluxes, etc.), we used the 70% footprint contour line that corresponds to an average footprint of 13042 m$^2$ of our salt marsh area of interest (Fig. 2). A land-use map was also created (Fig. 2) from geo-referenced IGN BD orthogonal images with a resolution of 20 cm (2019) using ArcGIS 10.2 (ESRI, Redlands, California, USA). The spatial analysis tool of ArcGIS 10.2 was used to perform an unsupervised classification of the BD orthogonal images. We checked the resulting map by selecting 20 random
locations within the footprint of the studied salt marsh and compared their land use on the ground and on the map.

### 2.4. EC data processing and quality control

Raw EC data measured at high-frequency were processed following Aubinet et al. (2000) with the EddyPro software. First, different correcting steps were applied to our raw data according to the procedures given by Vickers and Mahrt (1997)
and Polsenaere et al. (2012) for intertidal systems: (1) unit conversion to check that the units for instantaneous data are appropriate and consistent to avoid any errors in the calculation and correction of $CO_2$ fluxes, (2) despiking to remove outliers in the instantaneous data from the anemometer and gas analyser due to electronic and physical noise and replaced the detected spikes with a linear interpolation of the neighbouring values, (3) amplitude resolution to identify situations in which the signal variance is too low with respect to the instrumental resolution, (4) double coordinate rotation to align the x-
axis of the anemometer to the current mean streamlines, nullifying the vertical and cross-wind components, (5) time delay removal by detecting discontinuities and time shifts in the signal acquisition from the anemometer and gas analyser, (6) detrending with removal of short-term linear trends to suppress the impact of low-frequency air movements and (7) performing the Webb-Pearman-Leuning (WPL) correction to take into account the effects of temperature and water vapour fluctuations on the measured fluctuations in the $CO_2$ and $H_2O$ densities (Burba, 2021). The turbulent fluctuations of $CO_2$
fluxes were calculated with EddyPro using the linear detrending method (Gash and Culf, 1996) which involves calculating deviations from around any linear trend evaluated (i.e. over the whole flux averaged period). High-frequency $CO_2$ fluxes




were processed and averaged over intervals of 10 min. (shorter than in terrestrial ecosystems) to detect fast NEE variations with the tide at our site (Polsenaere et al., 2012; Van Dam et al., 2021).

A strict quality control was applied on EddyPro processed $CO_2$ flux data to remove bad data related to instrument
malfunctions, processing and mathematical artefacts, ambient conditions that do not satisfy the EC method, wind that is not from the footprint, and precipitation conditions (Burba, 2021). Processed data were screened using tests for steady state and turbulent conditions (Foken and Wichura, 1996; Foken et al., 2004; Göckede et al., 2004). If the signal to noise ratio of the EC150 gas analyser was less than 0.7 and/or the percentage of high-frequency missing values over 10 min. exceeds 10% (i.e. data absent in the raw data file or removed through the quality screening procedures), no flux was calculated. This choice
was the best compromise between removing poor-quality data and keeping as much of measured $CO_2$ flux data as possible (data and associated test not shown). Then, we used the method of Papale et al. (2006) to detect and remove outliers in the 10-min. flux data. The median and median absolute deviation (MAD) were calculated over a two-week window separating daytime and night-time periods. Data above 5.2×MAD were removed. After all post-processing and quality controls, 18.3% of the EC data were removed and gap-filled through a machine learning approach to obtain continuous flux data in 2020.


### 2.5. Flux gap filling and statistic tools

A random forest model was calculated to gap-fill our EC dataset. Random forest is a supervised machine learning technique proposed by Breiman (2001) that can model a non-linear relationship with no assumption about the underlying distribution of the data population. This method has been shown to be particularly suited to gap-fill EC data (Kim et al.,
2020; Cui et al., 2021). Random forest builds multiple decision trees, each of which are based on a bootstrap aggregated data sample (i.e. bagging of the EC data) and a random subset of predictors (i.e. the selected environmental data). The random forest model was built with environmental predictors that have been identified in the literature to control $CO_2$ fluxes in salt marshes and which were available during the gaps (PAR, air temperature, water level height and relative humidity) and with measurements recorded between 2019 and 2020. Each random forest model was built from a trained bagging ensemble of
400 randomly generated decision trees (Kim et al., 2020) with the "randomForest" package in the R software (Liaw and Wiener, 2002). Each tree was trained from bagged samples including 70% of the initial dataset. The remaining 30% of the data were used to estimate the fit of each random forest model. The model used was then able to explain 88% of the variability in the test data. Daytime data were better explained than night-time ones (59% *vs.* 38%), with light being the main parameter of the model. Using a partial dependence analysis and an ondelette analysis, we concluded that the relationships
and temporal dynamics modelled were sufficiently consistent to fill the gaps in our dataset. However, the model reduced the influence of certain variables on $CO_2$ flux during high PAR levels (> 1000 µmol m$^{-2}$ s$^{-1}$). This observation is common for random forest models, as they show poor results for extreme values. Other models such as artificial neural networks were also tested but showed poorer results.





For all measured variables, the high-frequency data (i.e. 10 min.) did not follow a normal distribution (Shapiro-Wilk

tests, p < 0.05). Non-parametric comparison tests such as the Mann-Whitney and Kruskal-Wallis were carried out with a

0.05 level of significance. To assess the influence of meteorological and hydrological drivers on NEE fluxes, we performed a

pairwise Spearman's correlation analysis on the 10-min. values and monthly mean values ("cor function" in R).

### 2.6. Temporal analysis of NEE fluxes and partitioning

Over the year 2020, temporal variations in NEE fluxes were studied at the seasonal and diurnal/tidal scales. Seasons

were defined based on calendar dates: the winter period from 01/01/2020 to 19/03/2020 and from 21/12/2020 to 31/12/2020,

the spring period from 20/03/2020 to 19/06/2020, the summer period from 20/06/2020 to 21/09/2020 and the fall period

from 22/09/2020 to 20/12/2020. Daytime and night-time were separated into PAR > 10 and PAR ≤ 10 µmol m$^{-2}$ s$^{-1}$,

respectively, and for the metabolic flux analysis, five PAR groups were chosen (0 < PAR ≤ 10, 10 < PAR ≤ 500, 500 < PAR

≤ 1000, 1000 < PAR ≤ 1500 and 1500 < PAR ≤ 2000 µmol m$^{-2}$ s$^{-1}$). Water heights (Hw) measured at one location over the

marsh (Fig. 2) relative to the mean sea level were used to distinguish emersion (Hw = 0 m at low tide) and immersion (Hw >

0 m at high tide) situations and thus, the influence of tides on NEE fluxes. However, in some situations based on the tide

(neap tides), due to meteorology influence (wind direction, atmospheric pressure) and the local altimetry heterogeneity, our

one-location Hw measurements could not accurately account for the whole spatial emersion and immersion of the marsh in

the EC footprint. When coastal waters begin to fill the channel and then overflow over the marsh (from 0.5 h in spring tides

to 2.5 h in neap tides; data not shown), the SSW sector (Fig. 2) was first immerged and a non-zero Hw value was measured.

However, although some marsh sectors were immerged at the same time, others were still emerged. Mud areas (lower levels)

were quickly immerged from Hw > 0 m (south) whereas the whole marsh immersion (muds and plants) only occurred 0.75 h

later from Hw > 1.0 m at high tide during spring tide. Conversely, at neap tide, this footprint immersion *vs* emersion marsh

heterogeneity could still be present even at high tide due to insufficient water levels. Although, a digital field model for

water heights could not be performed in 2020 to have a better spatial representation of the immersion/emersion footprint, all

these important considerations were considered in our computations and analysis in this study.

To study ecosystem metabolism related to photosynthesis and respiration processes, NEE fluxes (i.e. net vertical $CO_2$

exchanges measured by EC) were partitioned into gross primary production (GPP) and ecosystem respiration ($R_{eco}$),

respectively. The net ecosystem production (NEP), calculated as the difference between GPP and $R_{eco}$ (Eq. 2), allows to

describe the ability of an ecosystem to consume $CO_2$ and to produce OM (Gattuso et al., 1998). During marsh emersion,

NEE fluxes occur at the soil-atmosphere interface involving only benthic NEP (or marsh NEP) resulting in NEE = NEP.

During marsh immersion, NEE fluxes are the result of benthic NEP, planktonic NEP and lateral C exchanges by tides

thereby making it more difficult to study the marsh metabolism (Polsenaere et al., 2012). Negatives NEE and NEP values

indicated a $CO_2$ uptake by the marsh and positives values indicated a $CO_2$ source into the atmosphere. GPP was expressed in



negative values and $R_{eco}$ was expressed in positive values. In this study, NEP was calculated according to the following equation (Eq. 2) using the model of Kowalski et al. (2003):

$$NEP = GPP - R_{eco} = \frac{a1\,PAR}{a2+PAR} - R_{eco} \tag{2}$$

where $a_1$ is the maximal photosynthetic $CO_2$ uptake at light saturation (µmol $CO_2$ m$^{-2}$ s$^{-1}$) and $a_2$ is the PAR at half of the maximal photosynthetic $CO_2$ uptake (µmol photon m$^{-2}$ s$^{-1}$). The $a_1/a_2$ ratio corresponds to photosynthetic efficiency (Kowalski et al., 2003). $R_{eco}$ was calculated as follows (Eq. 3) according to Wei et al. (2020b):

$$R_{eco} = R_0 \exp(b\text{Ta}) \tag{3}$$

where $R_{eco}$ is the night-time ecosystem respiration (µmol $CO_2$ m$^{-2}$ s$^{-1}$), $R_0$ is the ecosystem respiration rate at 0°C (µmol $CO_2$ m$^{-2}$ s$^{-1}$), Ta is the measured air temperature (°C) and $b$ is a response coefficient of the temperature variation (Wei et al., 2020b).

All parameters used for NEE flux partitioning ($a_1$, $a_2$, $R_0$ and $b$; Eqs. 2 and 3) were estimated by the least square method ("minpack.lm" package in R) only during emersion periods at the monthly scale to better take into account the temporal variability of the coefficients. For each month, $R_0$ and $b$ were estimated during night-time emersion periods where NEE fluxes correspond to $R_{eco}$ fluxes (Wei et al., 2020b). Then, $a_1$ and $a_2$ were estimated during daytime emersion periods using $R_0$ and $b$ where NEE fluxes correspond to NEP fluxes (Kowalski et al., 2003). NEP fluxes (marsh metabolic fluxes without tidal influence) were calculated for each PAR and Ta value measured at a 10-min. frequency over the year using the monthly parameters calculated for the partitioning. As our ecosystem had a low phenological variation (Table A.1), we concluded that a monthly time step for the coefficient estimation was sufficient to answer our study objectives. During emersion periods, monthly net C balances (i.e. budgets) of measured NEE and estimated NEP were very similar as well as the monthly mean fluxes (Table A.2), confirming the correct NEE flux partitioning calculations done in this study.

## 3. Results

### 3.1. Habitat covering of the footprint

Within the EC footprint, halophile marsh vegetation (66%) composed of *Halimione portulacoides*, *Spartina maritima* and *Suaeda vera* mainly dominated when muds and channels only accounted for 27 and 7%, respectively (Fig. 2). The area occupied by *S. vera*, crossing the EC footprint from WNW to ESE (Table 1), corresponded to the highest marsh area that was partly immerged only during the highest tidal amplitudes (Fig. 2). *H. portulacoides* and *S. maritima* occupied in majority the NNE (70%), SSE (69%), WSW (68%) and SSW (67%) wind sectors. On the contrary, mud habitats mostly covered the NNW sector, where the lowest vegetation cover was found (Table 1 and Fig. 2). The highest channel area was found in the SSW sector (Table 1 and Fig. 2).





**Table 1: Bossys perdus marsh habitat (percentages % in bold and assocatied surface area m² in brackets) within each 45° wind sector in the corresponding footprint areas (Fig. 2) and the whole averaged footprint for the year 2020 (13042 m², 70% countour line). *Negligible surfaces on the total area of the sector.**


| Wind sectors | | *Halimione portulacoides* | *Spartina maritima* | *Suaeda vera* | Muds | Channels |
|---|---|---|---|---|---|---|
| NNE | 0-45 | **48** (850) | **22** (390) | **1*** (9) | **22** (386) | **8** (150) |
| ENE | 45-90 | **31** (590) | **26** (492) | **1** (22) | **37** (704) | **4** (80) |
| ESE | 90-135 | **37** (335) | **21** (190) | **31** (288) | **9** (82) | **2** (22) |
| SSE | 135-180 | **60** (803) | **9** (124) | **0*** (4) | **21** (275) | **8** (113) |
| SSW | 180-225 | **48** (734) | **19** (283) | **0*** (2) | **8** (122) | **25** (388) |
| WSW | 225-270 | **33** (689) | **35** (745) | **0*** (6) | **25** (530) | **6** (132) |
| WNW | 270-315 | **30** (580) | **11** (216) | **29** (570) | **30** (588) | **0** (0) |
| NNW | 315-360 | **16** (249) | **26** (401) | **2** (31) | **56** (867) | **0** (0) |
| Total footprint (70% contour line) | | **37** (4830) | **22** (2841) | **7** (932) | **27** (3554) | **7** (885) |

### 3.2. Seasonal variations of environmental conditions and NEE fluxes

Over the year 2020, the full seasonal range in solar radiation was measured (Fig. 3-A) with an increase in daytime PAR
from winter (lowest light season) to summer (brightest season). A similar seasonal pattern was recorded for air temperatures
(Fig. 3-B) with Ta values ranging from 1.5°C in winter (coldest season) to 33.6°C in summer (warmest season). On average,
the winter and fall seasons were the wettest (RH > 82%), associated with the lowest vapor pressure deficit (VPD) values
whereas spring and summer were the driest ones (RH < 75%), associated with the highest VPD values (Fig. 3-B). Indeed, the
highest and lowest cumulative rainfalls were recorded in fall (342 mm) and summer (62 mm), respectively. The highest
mean seasonal wind speed was measured in winter (4.9 ± 2.3 m s⁻¹) with maximal speeds up to 13 m s⁻¹ (Fig. 3-C). Winds
came in majority from the SSW-WSW sectors both in winter (55%) and summer (41%) and from the NNE-ENE sectors both
in spring (51%) and fall (31%). Tidal activities reflected the typical hydrological conditions of the Atlantic coasts with a bi-
weekly succession of spring tides and neap tides (Fig. 3-D). Water heights (Hw) strongly varied according to tidal
amplitudes with a maximal Hw of 1.4 m during neap tides and 2.0 m during spring tides (overall annual mean of 0.6 ± 0.4 m;
Fig. 3-D). At the annual scale, 25.5% of the EC data were measured when the salt marsh was immerged through variable
immersion durations and water heights (Table 2). On average, the daily immersion durations ranged between 5.7 h d⁻¹ in



winter (23.7% of the EC data) and 6.5 h d$^{-1}$ in fall (28% of the EC data). In winter, the EC data during immersion were split into 19% for 0 < Hw < 1 m and 4.7% for 1 < Hw < 2 m whereas in fall, these latter were split into 20% for 0 < Hw < 1 m and 8% for 1 < Hw < 2 m. In summer, the lowest marsh immersion was measured with no Hw value higher than 1.5 m (Table 2).

**Table 2: Emersion and immersion periods (percentage % in bold) at the studied salt marsh for four water height ranges of 0.5 m over the year 2020 and at the seasonal scale. In brackets, the emersion and immersion durations in hour per day (24-hour, h d$^{-1}$) were calculated.**

|  | Emersion | | | Immersion | |
| --- | --- | --- | --- | --- | --- |
|  | Hw = 0 | 0 < Hw < 0.5 | 0.5 < Hw < 1 | 1 < Hw < 1.5 | 1.5 < Hw < 2 |
| Year 2020 | **74.5** (17.9) | **12.4** (2.9) | **8.7** (2.1) | **3.6** (0.9) | **0.8** (0.2) |
| Winter | **76.3** (18.0) | **10.4** (2.5) | **8.6** (2.0) | **3.6** (0.9) | **1.1** (0.3) |
| Spring | **74.5** (18.0) | **13.7** (3.2) | **8.2** (2.0) | **3.0** (0.7) | **0.6** (0.1) |
| Summer | **75.1** (18.5) | **17.1** (4.2) | **5.9** (1.6) | **1.3** (0.3) | **0.0** (0.0) |
| Fall | **72.0** (17.0) | **8.5** (1.9) | **11.5** (2.7) | **6.4** (1.5) | **1.6** (0.4) |

The annual mean NEE value was -1.27 ± 3.48 µmol m$^{-2}$ s$^{-1}$ with strong temporal variabilities recorded over both long and short timescales (Fig. 3-E). Significant NEE variations were highlighted between each studied season (Kruskal-Wallis test, p < 0.001), where the highest and lowest atmospheric CO$_2$ sinks were recorded in spring (-1.93 ± 3.84 µmol m$^{-2}$ s$^{-1}$) and fall (-0.59 ± 2.83 µmol m$^{-2}$ s$^{-1}$), respectively (Fig. 4). NEE flux partitioning gave an annual mean NEP value of -1.28 ± 3.16 µmol m$^{-2}$ s$^{-1}$, ranging from -2.00 ± 3.49 µmol m$^{-2}$ s$^{-1}$ in spring to -0.53 ± 2.51 µmol m$^{-2}$ s$^{-1}$ in fall. On average, in winter and fall, the measured NEE values were lower and more negative than the estimated NEP values whereas in spring and summer, the opposite trend was recorded (Fig. 4). Contrary to NEE and NEP, the highest seasonal values of GPP and R$_{eco}$ were estimated in summer whereas the lowest seasonal values were estimated in winter (Fig. 4). The highest and lowest photosynthetic efficiencies (a$_1$/a$_2$ ratio) were found in winter (-2.08 10$^{-2}$) and in summer (-1.36 10$^{-2}$), respectively.





**Figure 3: Net ecosystem exchanges and associated environmental parameters measured every 10 minutes over the year 2020. The measured environmental parameters include (A) the photosynthetically active radiation (PAR, µmol m$^{-2}$ s$^{-1}$), (B) air temperature (Ta, °C), vapor pressure deficit (VPD), (C) wind speed (m s$^{-1}$), (D) water height (Hw, m), water temperature (Tw, °C) and (E) the net ecosystem exchanges (NEE, µmol $CO_2$ m$^{-2}$ s$^{-1}$) computed from the 20 Hz atmospheric $CO_2$ and wind speed measurements with the EddyPro software. The red line in Fig. 3-E is the moving average of NEE (daily mean). Seasons are delimited by vertical lines.**



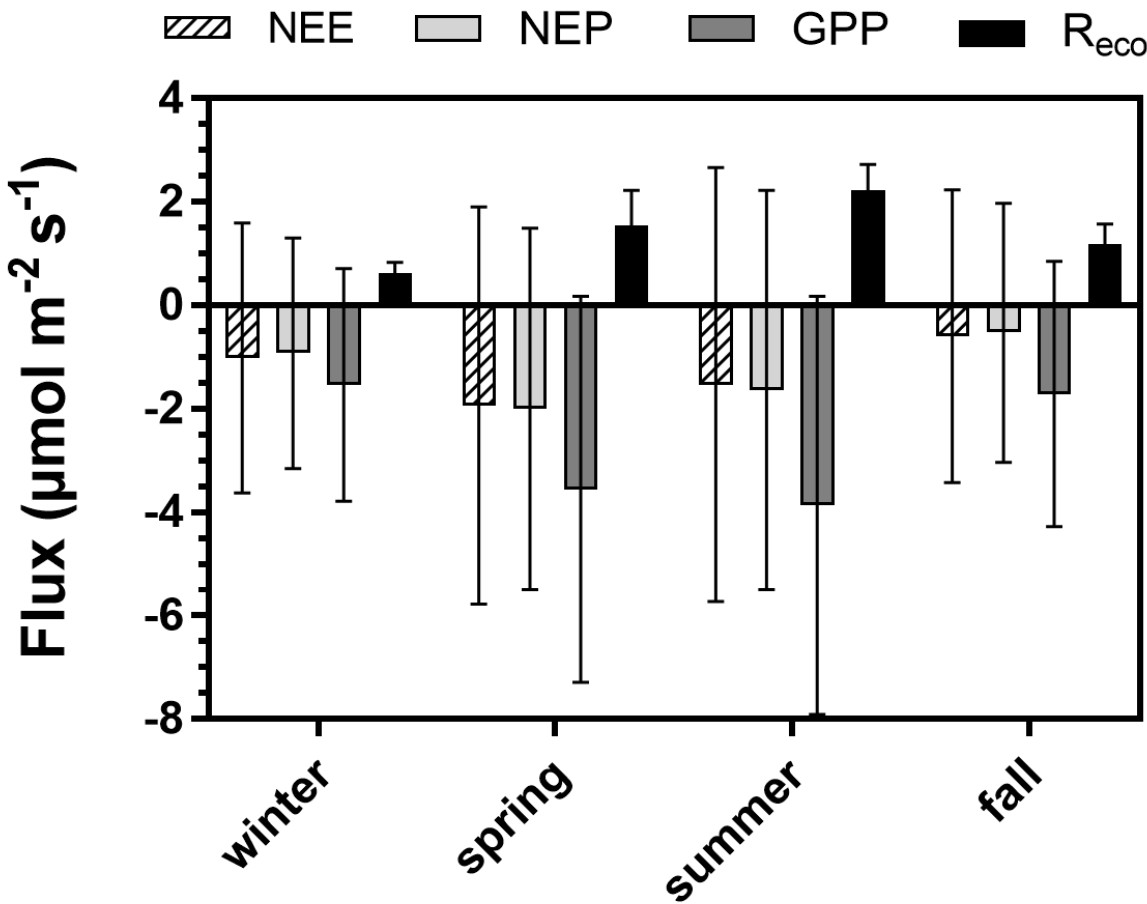

**Figure 4: Seasonal variations (means ± SD) of the measured NEE, estimated NEP, estimated GPP and estimated R_eco (µmol m$^{-2}$ s$^{-1}$) values recorded over the year 2020. NEE: net ecosystem exchange, NEP: net ecosystem production, GPP: gross primary production, R_eco: ecosystem respiration. The NEE fluxes were partitioned into GPP and R_eco according to Kowalski et al. (2003).**

### 3.3. Environmental parameter and NEE flux variations at diurnal and tidal scales

At each season, significant diurnal differences in NEE fluxes were highlighted (Mann-Whitney tests, $p < 0.05$) with, on average, an atmospheric $CO_2$ sink during daytime and an atmospheric $CO_2$ source during night-time, irrespective of emersion or immersion periods (Table 3). For instance, in spring, NEE means were -3.93 ± 3.72 and 1.06 ± 1.09 µmol m$^{-2}$ s$^{-1}$





during daytime and night-time, respectively (Fig. 5-B). Over all seasons, similar diurnal variations in measured NEE and
estimated NEP were recorded with, on average, a rapid increase in $CO_2$ uptake during the morning up to the middle of the
day (high PAR and low $Ta$) and then, a decrease in $CO_2$ uptake during the afternoon (high PAR and high $Ta$) to become a
$CO_2$ source during night-time (Fig. 5). On average, during the afternoon, the GPP decreases and $R_{eco}$ increases explained the
measured decrease in $CO_2$ uptake. For each season, the highest $CO_2$ uptakes were measured during emersion between 12:00
and 13:00 (maximal PAR levels), with the latter increasing from winter (-4.84 ± 2.87 µmol m$^{-2}$ s$^{-1}$) to spring-summer (-6.94
± 2.80 µmol m$^{-2}$ s$^{-1}$; Fig. 5).

**Table 3: Diurnal/tidal variations (means ± SD in bold) of NEE fluxes (µmol $CO_2$ m$^{-2}$ s$^{-1}$) during each season in 2020. The associated ranges (min/max) are indicated in brackets. Daytime and night-time periods were separated into PAR > 10 and PAR ≤ 10 µmol m$^{-2}$ s$^{-1}$, respectively, whereas emersion and immersion periods were separated into Hw = 0 m and Hw > 0 m, respectively.**

|  | Daytime Emersion | Night-time Emersion | Daytime Immersion | Night-time Immersion | Seasonal |
|---|---|---|---|---|---|
| Winter | **-3.15 ± 2.96** (-19.55/10.73) | **0.61 ± 0.86** (-4.80/5.40) | **-2.03 ± 2.30** (-16.06/6.49) | **-0.10 ± 0.99** (-5.31/3.34) | **-1.01 ± 2.61** (-19.55/10.73) |
| Spring | **-4.39 ± 3.76** (-25.67/19.09) | **1.25 ± 0.98** (-4.54/7.01) | **-2.59 ± 3.24** (-29.68/17.62) | **0.51 ± 1.22** (-4.60/6.04) | **-1.93 ± 3.84** (-29.68/19.09) |
| Summer | **-4.42 ± 3.88** (-23.71/18.07) | **2.11 ± 1.34** (-5.93/9.25) | **-2.22 ± 3.26** (-25.23/13.01) | **1.18 ± 1.44** (-4.86/9.36) | **-1.53 ± 4.19** (-25.23/18.07) |
| Fall | **-3.00 ± 3.32** (-21.54/17.74) | **1.12 ± 1.03** (-4.19/6.09) | **-1.53 ± 2.60** (-18.15/18.21) | **0.29 ± 1.07** (-3.97/5.50) | **-0.59 ± 2.83** (-21.54/18.21) |

At each season, the tidal rhythm could strongly disrupt NEE fluxes with, in general, no change in the marsh metabolism
status (sink/source). During daytime, significant lower $CO_2$ uptakes were recorded during immersion than during emersion
(Mann-Whitney tests, p < 0.05) when marsh plants were mostly immerged in tidal waters and during night-time, a similar
tidal pattern was recorded for $CO_2$ emissions (Mann-Whitney tests, p < 0.05; Table 3). For instance, in spring, NEE means
were -4.39 ± 3.76 and -2.59 ± 3.24 µmol m$^{-2}$ s$^{-1}$ during daytime emersion and daytime immersion, respectively, and were
1.25 ± 0.98 and 0.51 ± 1.22 µmol m$^{-2}$ s$^{-1}$ during night-time emersion and night-time immersion, respectively. In winter,
during night-time immersion (Fig. A.3) and, at particular moments (i.e. 143 hours over 55 days associated with a mean Hw
of 0.80 m), a weak $CO_2$ sink was recorded (-0.82 ± 0.91 µmol m$^{-2}$ s$^{-1}$) with a maximal uptake of -5.31 µmol m$^{-2}$ s$^{-1}$ (Table 3).





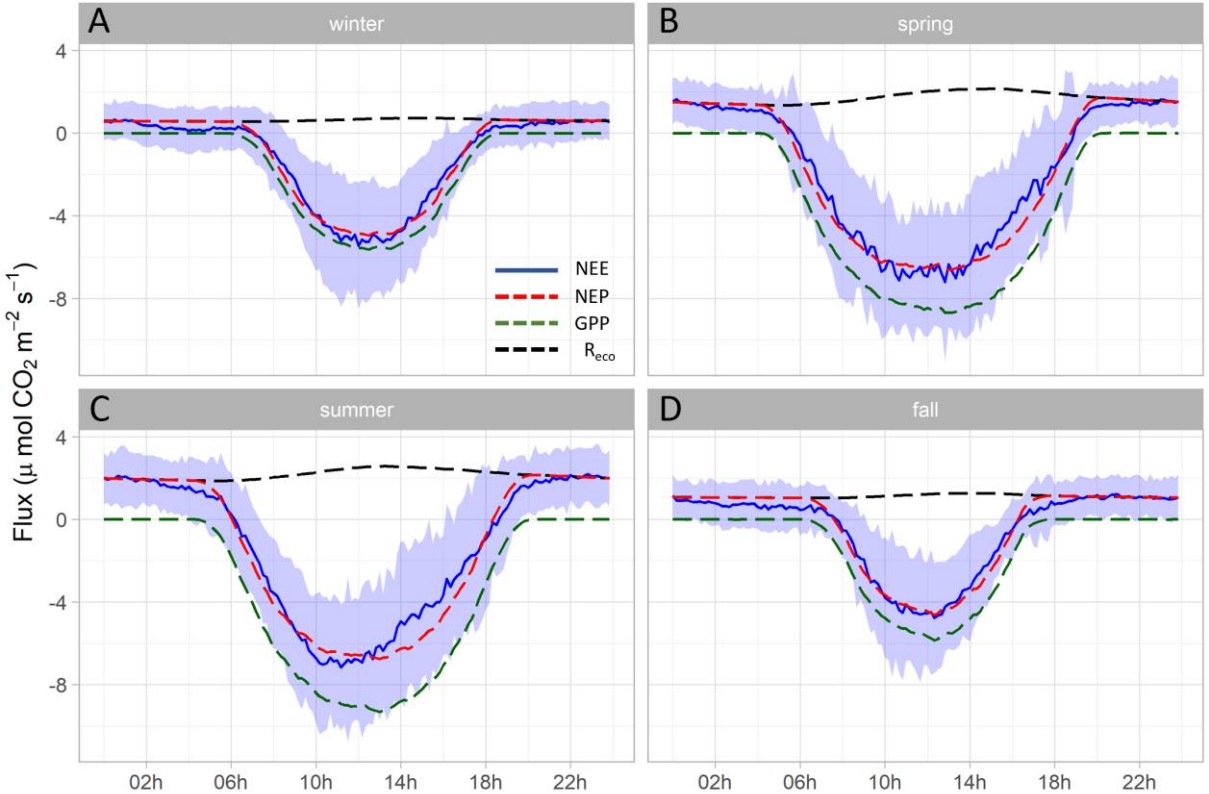

**Figure 5: Hourly plots of the NEE, NEP, GPP and R$_{eco}$ diurnal variations obtained every 10 minutes in winter (A), spring (B), summer (C) and fall (D) over the year 2020. NEE averages are represented by blue solid lines whereas standard deviations are represented by blue areas; the NEP, GPP and R$_{eco}$ averages are represented by red, green and black dotted lines, respectively. The NEE fluxes were partitioned into GPP and R$_{eco}$ according to Kowalski et al. (2003) using monthly coefficients (see the M&M section). Night-time periods correspond to GPP = 0 μmol m$^{-2}$ s$^{-1}$ and NEP = R$_{eco}$. All values are in μmol CO$_2$ m$^{-2}$ s$^{-1}$.**

## 3.4. Influence of environmental drivers on temporal NEE variations

Over the year, NEE fluxes were significantly controlled by solar radiations and air temperatures at the multiple timescales studied, thereby favouring marsh CO$_2$ uptake. During daytime (PAR > 10 μmol m$^{-2}$ s$^{-1}$), PAR and Ta displayed the strongest negative correlations with NEE at both the monthly scale (-0.87 and -0.65, respectively, $p < 0.05$) and the 10-minute scale (-0.77 and -0.21, respectively, $p < 0.05$). The highest and lowest correlations between NEE and PAR were recorded for $10 < PAR \leq 500$ and for $1500 < PAR \leq 2000$ μmol m$^{-2}$ s$^{-1}$, respectively, confirming the rapid increase or decrease in CO$_2$ uptake for low PAR values (Fig. 6-A). For PAR $\leq 500$ μmol m$^{-2}$ s$^{-1}$ (night, dawn, dusk or low light days), relative humidity (RH) did not influence NEE whereas from PAR > 500 μmol m$^{-2}$ s$^{-1}$, RH was negatively correlated with NEE and the wettest periods favoured CO$_2$ uptake (Fig. 6-B). During night-time and daytime, air temperature (Ta) was





positively and negatively correlated with NEE (0.54 and -0.21, respectively, p < 0.05). However, from PAR > 500 µmol m$^{-2}$
s$^{-1}$, high Ta values (> 20°C) decreased $CO_2$ uptake for all PAR levels (Fig. 6-C). Water temperature (Tw) did not influence
NEE during immersion (Fig. 6-D). Indeed, for PAR > 500 µmol m$^{-2}$ s$^{-1}$ and Hw > 0.5 m, no significant relationships was
found between NEE and Tw (n = 1215; p = 0.26). For wind directions, the highest $CO_2$ uptake and $CO_2$ emission were
measured, on average, over the SSE and NNW sectors, respectively (Fig. 6-E). High wind speeds (> 7 m s$^{-1}$) increased $CO_2$
uptake whereas slow wind speeds did not influence NEE (Fig. 6-F).


**Figure 6: Diurnal variations of NEE (µmol $CO_2$ m$^{-2}$ s$^{-1}$) obtained every 10 minutes according to different variables within five PAR groups: 0-10 (night-time), 10-500, 500-1000, 1000-1500 and 1500-2000 µmol m$^{-2}$ s$^{-1}$. PAR (µmol m$^{-2}$ s$^{-1}$; A), relative humidity (%, B), air temperature (°C, C), water temperature (°C, D), wind direction (°, E) and wind speed (m s$^{-1}$, F). NEE fluxes are**
**averaged after separating each variable into five classes and the coloured area is the standard error at the mean.**



The tidal rhythm strongly influenced NEE during immersion depending on the measured water heights (Hw) and PAR levels (Figs. 7 and A.2). Over the year, NEE were positively correlated with Hw during daytime but negatively correlated during night-time (Fig. 7). More precisely, night-time immersion strongly reduced $CO_2$ emissions and even led to a switch from source to sink of atmospheric $CO_2$ from Hw > 0.4 m in winter (Fig. 7-A), Hw > 0.7 m in spring (Fig. 7-B), Hw > 1.4 m in summer (Fig. 7-C) and Hw > 1 m in fall (Fig. 7-D), on average. For low daytime PAR levels (PAR < 500 µmol m$^{-2}$ s$^{-1}$), immersion only slightly reduced $CO_2$ uptake (Fig. 7-C). On the contrary, for higher daytime PAR levels (PAR > 500 µmol m$^{-2}$ s$^{-1}$), immersion strongly reduced $CO_2$ uptake, especially in summer, where the lowest $CO_2$ sink was reached for 1 < Hw < 1.5 m, irrespective of the PAR levels (Fig. 7-C). For instance, in summer, daytime NEE means were -2.71 ± 3.48 and -0.16 ± 0.98 µmol m$^{-2}$ s$^{-1}$ for 0 < Hw < 0.5 m and 1 < Hw < 1.5 m, respectively. At each season, from Hw > 1.3 m, the influence of PAR levels on diurnal NEE variations remained low (Fig. 7), reducing the solar radiation contribution on $CO_2$ uptake. However, during certain daytime immersion periods, incoming tides can temporally favour $CO_2$ uptake (Fig. 7). For instance, in spring, during daytime immersion (1500 < PAR ≤ 2000 µmol m$^{-2}$ s$^{-1}$), NEE means were -2.89 ± 2.53 and -5.59 ± 2.83 µmol m$^{-2}$ s$^{-1}$ for Hw = 0.9 m and Hw = 1.1 m, respectively (Fig. 7-B).

### 3.5. Annual Carbon budgets

Over the year, the annual NEE value was -483.6 g C m$^{-2}$ yr$^{-1}$, associated with an average immersion duration of 6.1 h d$^{-1}$. Simultaneously, GPP and $R_{eco}$ absorbed and emitted 1019.4 and 533.2 g C m$^{-2}$ yr$^{-1}$, respectively, resulting in an annual NEP value similar to the NEE value (Fig. 8). At the seasonal scale, the highest $CO_2$ uptakes occurred in spring and summer, associated with the lowest marsh immersion levels, and the lowest $CO_2$ uptakes occurred in winter and fall, associated with the highest marsh immersion levels (Tables 2 and 4). In winter and fall, when the daytime immersion periods were the shortest, net C balances from measured NEE gave higher values than values calculated from estimated NEP (+7.9 and +6.2 g C m$^{-2}$, respectively; Table 4). Conversely, in spring and summer when the daytime immersion periods were the longest, the opposite pattern was observed between measured NEE values and estimated NEP values (-7.3 and -9.9 g C m$^{-2}$, respectively; Table 4).





**Figure 7: Diurnal variations of NEE (µmol CO₂ m⁻² s⁻¹) obtained every 10 minutes according to water height (Hw, m) within five PAR groups (see captions in Fig. 6) in winter (A), spring (B), summer (C) and fall (D). NEE values were averaged every 0.1 m. The coloured areas represent the standard error of the mean.**





**Table 4: Net seasonal carbon balances for the measured NEE and estimated NEP values (g C m$^{-2}$). Corresponding seasonal percentages (%) of marsh immersion and daytime marsh immersion are indicated. For more information on NEE flux partitioning into NEP, see the M&M section.**

|  | Cumulative NEE (g C m$^{-2}$) | Cumulative NEP (g C m$^{-2}$) | NEE – NEP (g C m$^{-2}$) | Immersion time (%) | Daytime immersion time (%) |
|---|---|---|---|---|---|
| Year 2020 | 483.6 | 485.9 | -2.3 | 25.5 | 52.2 |
| Winter | 94.4 | 86.5 | 7.9 | 23.7 | 41.5 |
| Spring | 184.5 | 191.8 | -7.3 | 25.5 | 63.4 |
| Summer | 149.3 | 159.2 | -9.9 | 24.9 | 64.5 |
| Fall | 55.5 | 49.3 | 6.2 | 27.9 | 39.5 |

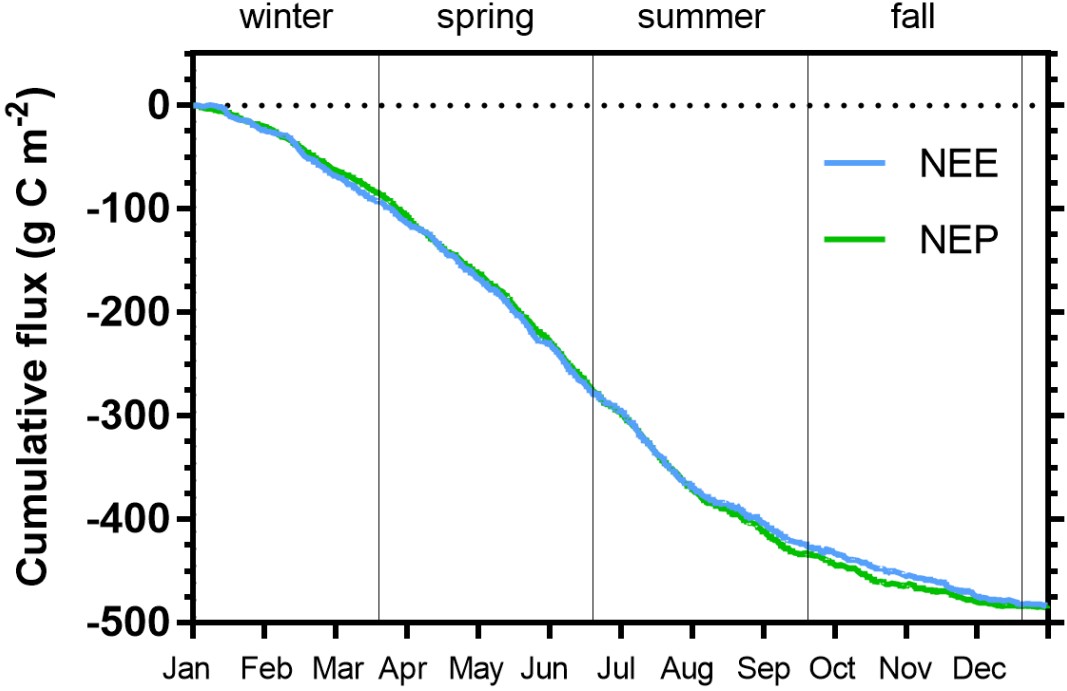

**Figure 8: Cumulative fluxes (g C m$^{-2}$) of the measured NEE (in blue) and estimated NEP (in green) throughout the year 2020. Vertical lines are used to delimit the four seasons. NEE fluxes correspond to net vertical CO$_2$ exchanges measured by EC whereas NEP fluxes correspond to net vertical CO$_2$ exchanges estimated from NEE partitioning at the benthic interface only, without any tidal influence.**





**Table 5: Comparison of the annual NEE budget (g C m⁻² yr⁻¹) using EC measurements across the salt, brackish and freshwater marshes of the coastal zone.**

| Study sites | Locations | Annual NEE budgets (g C m$^{-2}$ yr$^{-1}$) | References |
|---|---|---|---|
| Tidal salt marsh[*] | Fier d'Ars tidal estuary, France | -483 | This study |
| Tidal salt marsh[*] | Virginia, USA | -130[a] | Kathilankal et al., 2008 |
| Urban tidal marsh[*] | Hudson-Raritan estuary, New-Jersey, USA | From +894 to -310 | Schäfer et al., 2014 |
| Restored salt marsh[*] | Hudson-Raritan estuary, New-Jersey, USA | -213 | Artigas et al., 2015 |
| Tidal salt marsh | Plum Island Sound estuary, Massachusetts, USA | From -104 to -233 (-176 ± 32)[b] | Forbrich et al., 2018 |
| Tidal salt marsh | Duplin River salt marsh-estuary, Georgia, USA | From -139 to -309 | Nahrawi, 2019 |
| Urban tidal wetlands | Hudson-Raritan estuary, New-Jersey, USA | -307[c] | Schäfer et al., 2019 |
| Brackish tidal marsh | San Francisco Bay, California, USA | -225 | Knox et al., 2018 |
| Brackish marsh | Louisiana, USA | 171 | Krauss et al., 2016 |
| Para-dominated subtropical marsh | Taiwan | -376 | Lee et al., 2015 |
| Reed-dominated marsh | Taiwan | -53 | Lee et al., 2015 |
| Freshwater marsh | Louisiana, USA | -337 | Krauss et al., 2016 |
| Freshwater wetland | Everglades National Park, Florida, USA | From -91 to +3 (-21 ± 17)[d] | Zhao et al., 2019 |

[*]**Managed and protected marshes, [a]NEE budget during the growing season (from May to October 2007), [b]Mean of annual NEE budgets over a five-year period (from 2013 to 2017), [c]Annual NEE budget of three tidal marshes with different restoration**
**histories, [d]Mean of annual NEE budgets over a nine-year period (from 2008 to 2016).**



## 4. Discussion

### 4.1. Atmospheric CO$_2$ exchanges and associated methodological aspects

In the present EC study, the studied salt marsh absorbed 483 g C m$^{-2}$ yr$^{-1}$ from the atmosphere. This net C balance (i.e. budget) was lower than the values estimated for global tidal wetlands (1125 g C m$^{-2}$ yr$^{-1}$; Bauer et al., 2013) and for tidal

marshes on the U.S. Atlantic coast (775 g C m$^{-2}$ yr$^{-1}$; Wang et al., 2016) but similar to the C balance estimated by Alongi (2020) for global salt marshes (382 g C m$^{-2}$ yr$^{-1}$). Currently, an increasing number of EC measurements are being taken in salt marshes in order to obtain continuous NEE data series as well as to increase knowledge about the associated metabolic processes and fluxes for these tidal systems (Schäfer et al., 2014; Forbrich et al., 2018; Knox et al., 2018) (Table 5). The advantage of the atmospheric EC method in salt marshes is that it measures ecosystem-scale NEE fluxes at both the benthic

and aquatic interfaces to obtain high-resolution continuous NEE time series. The drawback to EC is a loss of data related to instrument malfunctions, processing and mathematical artefacts, ambient conditions that do not satisfy the requirements for the EC method as well as heavy precipitation for open-path IRGA (Burba, 2021) that needs to be gap-filled by machine learning methods for instance (Artigas et al., 2015; Forbrich et al., 2018; Knox et al., 2018; Schäfer et al., 2019). Here, a random forest model, based on the PAR, Ta, Hw and RH values, was used to estimate and gap fill the 18.3% missing EC

data over the year (Kim et al., 2020; Cui et al., 2021). Moreover, another methodological aspect is the partitioning of NEE fluxes into GPP and R$_{eco}$ to study the terrestrial (benthic) metabolism of the salt marsh related to photosynthesis and respiration processes from plants and soils (Forbrich and Giblin, 2015; Knox et al., 2018; Wei et al., 2020b). Here, parameter calculations performed at the monthly scale for the NEE flux partitioning (Kowalski et al., 2003; Wei et al., 2020b) were found to be the best option based on our present study objectives and due to the low phenological variation of the marsh

plants.

### 4.2. Influence of marsh CO$_2$ uptake and management practice

The EC technique confirmed the estimates of CO$_2$ sinks in salt marshes (Wang et al., 2016; Alongi, 2020) but also revealed strong NEE flux heterogeneities according to climatic conditions and anthropogenic influences (Herbst et al., 2013;

Schäfer et al., 2019). For instance, NEE measured in a natural salt marsh showed a net C uptake from the atmosphere with strong interannual variations in C balances (Table 5) mainly due to rainfall during the growing season for marsh plants (Forbrich et al., 2018). By comparison, in an urban tidal marsh, Schäfer et al. (2014) reported a higher interannual variability from 984 g C m$^{-2}$ in 2009 to -310 g C m$^{-2}$ in 2012 due to management practices and plant species (*P. australis* and *S. alterniflora* in 2009 and total elimination of *P. australis* in 2012; Table 5). In the same area, in another restored salt marsh in

which the *P. australis* monoculture was replaced by a high diversity of emergent marsh plants (*S. patens*, *S. cynosuroides*, *S. alterniflora* and *D. spicata*), a net CO$_2$ uptake was recorded (Table 5) which once again confirms the importance of land



management practices in marsh C balances (Artigas et al., 2015). In our studied salt marsh, the natural management for several decades has allowed for a return to the natural site hydrodynamics and the development of productive marsh halophytes, mainly composed of *H. portulacoides* and *S. maritima* (59% of the footprint area). However, past human activities and water management practices for salt farming have shaped the marsh typology (channel network, humps, dykes), producing a time-delayed immersion of plants and muds between high and low marsh areas during spring tides. During the year 2020, our rewilded salt marsh did uptake more C from the atmosphere mainly due to strong plant photosynthesis than the other salt, brackish and freshwater marshes reported in the literature (Table 5). However, the net C balances calculated using the EC approach are still too scarce to be able to take all annual and spatial variabilities of salt marshes into account. Based on biomass production measurements in salt marshes, Sousa et al. (2010) estimated that the NPP of *H. portulacoides* was 505 g C m$^{-2}$ yr$^{-1}$ whereas the NPP of *S. maritima* varied between 367 and 959 g C m$^{-2}$ yr$^{-1}$ depending on the chemical-physical characteristics and marsh maturity; these NPP values were similar to our net C balance. Moreover, the microphytobenthos that developed on mudflats and channels (34% of the footprint area) may also contribute to marsh benthic production during daytime emersion, as highlighted in our studied salt marsh where static chamber measurements performed in March 2023 at midday showed a $CO_2$ uptake to a non-vegetated mudflat (NEE mean of -2.92 µmol m$^{-2}$ s$^{-1}$; unpublished results) and confirmed in an estuarine wetland in China (Xi et al., 2019). On an intertidal flat (France), EC measurements even showed a higher daily benthic metabolism with microphytobenthos (1.72 g C m$^{-2}$ d$^{-1}$; September/October 2007) than with *Zostera noltii* (1.25 g C m$^{-2}$ d$^{-1}$; July and September 2008), confirming the high biological productivity of mudflats (Polsenaere et al., 2012).

### 4.3. Metabolism processes and controlling factors at multiple timescales

#### 4.3.1. Seasonal scale

The average monthly budgets from Forbrich et al. (2018) showed, at their site, a net $CO_2$ sink during the growing season for marsh plants from June to September and a net $CO_2$ source to the atmosphere during the rest of the year, indicating a strong seasonal variability in NEE. In urban salt marshes, the growing season was longer switching from source to sink in May (Schäfer et al., 2014; Artigas et al., 2015) and even in April in a brackish marsh (Knox et al., 2018). In our study, the salt marsh behaved as a net $CO_2$ sink throughout the year during both the growing and non-growing seasons with the highest and lowest C uptake from the atmosphere reached in July (73 g C m$^{-2}$) and December (9 g C m$^{-2}$), respectively (Fig. 8). The low $R_{eco}$ rates related to plant and soil respiration processes resulted in lower atmospheric $CO_2$ emissions in the studied salt marsh than in urban salt marshes (Artigas et al., 2015) and brackish marshes (Knox et al., 2018), thus allowing for a net $CO_2$ sink from winter to summer. Moreover, our low $R_{eco}$ is also likely linked to the low OM decomposition observed at our site, notably due to recalcitrant OM (Arnaud et al., submitted 2022). Furthermore, it is also important to better understand the





direct and indirect effects of meteorological conditions and tidal immersion on photosynthesis and respiration processes and the associated marsh C balances (Knox et al., 2018).

Our study showed the predominant role of PAR and Ta on NEE variations in the studied salt marsh as has already been highlighted elsewhere by Wei et al. (2020b). Our results on the NEE flux partitioning into GPP and $R_{eco}$ during emersion indicated that plant photosynthesis was mainly driven by light, while ecosystem respiration was mainly driven by temperature. At the seasonal scale, the strongest $CO_2$ sinks were measured during warm and bright periods such as spring and summer, which were responsible for 70% of the annual C uptake (Table 4). However, although the highest seasonal rate

of GPP was measured in summer during the brightest months, the simultaneously recorded high Ta values instead favoured ecosystem respiration producing a lower $CO_2$ uptake in summer than in spring (Table 4). For instance, in two urban salt marshes, the Ta values above 30°C reduced $CO_2$ uptake by increasing respiration and atmospheric $CO_2$ emissions (Schäfer et al., 2019). These two meteorological parameters controlled short- and long-term NEE variations, as confirmed in urban salt marshes where significant and strong pairwise correlations of NEE with net radiation and temperature were recorded on half

hourly, daily and monthly averages (Schäfer et al., 2019).

At the studied salt marsh, we showed a significant influence of RH and VPD on daytime NEE variations favouring $CO_2$ uptake for the highest RH values (from 80 to 100%). The lack of a significant relationship between NEE and RH at night indicated that humidity influenced plant photosynthesis, by decreasing VPD and stomata opening, rather than their respiration. In a similar tidal salt marsh, Forbrich et al. (2018) showed a link between rainfall and C budgets on interannual

variations in NEE, i.e. during the early growing season in spring, rainfall events produced a decrease in soil salinity and favoured $CO_2$ uptake through an increase in plant productivity. In a salt marsh in the Yellow River Delta, significant NEE increases and GPP decreases were recorded with high soil salinities during emersion using static chamber measurements (Wei et al., 2020a). High levels of soil salinity in salt marshes are a stressor for plants such as *Spartina spp.* and can lead to reduce biomass production by inhibiting nutrient and $CO_2$ uptake throughout stomatal closure (Morris, 1984; Hwang and

Morris, 1994). Thus, in our studied marsh, we believe that the increase in dryness periods, especially during the summertime, with a decrease in rainfall events could profoundly modify plant productivity and marsh C uptake. This was confirmed by a significant reduction in the $CO_2$ sink at the studied salt marsh with low RH and high Ta values.

### 4.3.2. Diurnal and tidal scale influences

High-frequency EC measurements demonstrate that diurnal variations in NEE fluxes were driven by light rather than air temperature (Xi et al., 2019; Wei et al., 2020b) with no significant time-delay recorded between NEE and PAR variations (Fig. A.3). At our studied site, the highest negative correlations between NEE and PAR were highlighted for low daytime PAR values, indicating that light increases during the morning strongly favoured $CO_2$ uptake mainly through plant photosynthesis up to the middle of the day. The high Ta and VPD values recorded during the afternoon favoured ecosystem





respiration rather than photosynthesis and, in turn, a reduction in net $CO_2$ uptake up to reach $CO_2$ emissions during night-time (Knox et al., 2018; Xi et al., 2019). In another tidal salt marsh, Kathilankal et al. (2008) confirmed the PAR importance on *Spartina* photosynthesis and diurnal NEE fluxes. In a restored salt marsh, EC measurements also showed that the time of day has a major influence on atmospheric $CO_2$ exchanges during the growing season, accounting for 49% of NEE variability (Artigas et al., 2015). Moreover, in some cases, soil respiration can also be controlled by PAR or photosynthesis at the

diurnal scale (Vargas et al., 2011; Jia et al., 2018; Mitra et al., 2019), once again highlighting the major role played by light in diurnal NEE variations (Kathilankal et al., 2008; Wei et al., 2020b).

At the daily scale, the intensity of atmospheric $CO_2$ exchanges and the metabolic status of the marsh (sink/source) were also significantly influenced by the tidal rhythm (Fig. 7). Tides produced a significant decrease in daytime $CO_2$ uptake with maximal reductions up to 90% for the highest tidal amplitudes. In a *S. alterniflora* salt marsh, a mean reduction of 46 ± 26%

was measured during immersion, although large $CO_2$ amounts were still assimilated at a reduced rate (Kathilankal et al., 2008). In some cases, daytime NEE fluxes could be completely suppressed during immersion in salt marshes (Moffett et al., 2010; Forbrich and Giblin, 2015; Wei et al., 2020a) and brackish marshes (Knox et al., 2018). This drop in $CO_2$ uptake could be related to a physiological stress for plants under tidal immersion conditions resulting in a reduction of the effective photosynthetic leaf area and photosynthesis rates (Kathilankal et al., 2008; Moffett et al., 2010). Moreover, the physical

barrier created by tidal waters could limit the $CO_2$ diffusion from waters to plants, thereby resulting in fewer $CO_2$ exchanges between the atmosphere and the benthic compartment (sediments, soil). Using chamber measurements at different tidal stages, Wei et al. (2020a) also highlighted the importance of water heights and marsh immersion levels in NEE variations and confirmed a significant GPP decrease during immersion. However, tidal effects on daytime NEE fluxes may be more variable depending on the immersion level of the marsh and the biogeochemistry state of the tidal waters. Indeed, during the

brightest periods in winter and spring, the temporary increases in $CO_2$ uptake recorded during incoming tides could be related to (1) an increase in the GPP of plants favoured by RH increases and Ta decreases due to tidal conditions and/or (2) tidal waters advected from the shelf that are undersaturated in $CO_2$ with respect to the atmosphere due to phytoplankton blooms (Mayen et al., in prep.). Moreover, when the salt marsh was fully immerged at high tide during spring tides, NEE fluxes were mostly controlled by ecosystem respiration or/and inorganic processes (carbonate and physicochemical pumps)

rather than by photosynthesis, as light was no longer a major control factor for $CO_2$ uptake in tidal waters.

During night-time, $CO_2$ emissions from the salt marsh were inhibited by tidal effects through a significant decrease in ecosystem respiration (Han et al., 2015; Knox et al., 2018; Wei et al., 2020a). The physical barrier formed by tidal waters limits the atmospheric $CO_2$ releases via respiration from plants and soils (Wei et al., 2020b). Moreover, saturation of surface soils in tidal waters during immersion could reduce oxygen availability in the soil and limit OM microbial decomposition

and $CO_2$ emissions through aerobic respiration (Nyman and DeLaune, 1991; Miller et al., 2001; Jimenez et al., 2012; Han et al., 2015). In our case, night-time $CO_2$ exchanges were reduced up to 100% (completely suppressed), sometimes even



causing a change in metabolic status of atmospheric $CO_2$ from source to sink, especially in winter when the $R_{eco}$ rates were the lowest (Fig. 7). The presence of tidal waters advected from the shelf during the night and $CO_2$ undersaturated with respect to the atmosphere due to previous phytoplankton production and/or $CaCO_2$ dissolution in the water column during 605 the day (Gattuso et al., 1999; Polsenaere et al., 2012), could induce a sink which may lead to a net uptake of $CO_2$ at night. The results of our study indicate that tidal NEE variations may be mainly related to the marsh immersion level, the PAR level and the time of the growing cycle of plants as reported in Nahrawi et al. (2020).

### 4.4. Salt marsh carbon budgets for future research perspectives

At the annual scale in 2020, the tidal rhythm did not significantly affect the net C balance of the studied salt marsh since similar annual measured NEE and estimated NEP values were recorded (Fig. 8). The loss of $CO_2$ uptake measured during daytime immersion due to a GPP decrease could be compensated by night-time immersion where $CO_2$ emissions and $R_{eco}$ were inhibited. However, strong temporal variabilities were measured, especially between the growing and non-growing seasons. In winter and fall, the salt marsh uptaked more C from the atmosphere with the tidal influence (measured NEE) than 615    without (estimated NEP), especially in December (+35.7%), November (+19.7%) and January (+15.4%), associated with the highest photosynthetic efficiencies. An opposite trend was observed in spring and summer with a reduction in net C uptake under tidal influence, especially in August (-16.9%) and September (-9.8%). This significant difference in the seasonal C balances could be mainly related to the photoperiod of immersion periods. We demonstrated that daytime immersion decreased $CO_2$ uptake, whereas night-time immersion decreased $CO_2$ emissions up to a change in metabolic status for the 620    highest immersion levels. Thus, during seasons where daytime immersion primarily occurs, such as spring and summer, the salt marsh uptaked less atmospheric $CO_2$ with the tidal influence, whereas seasons that mostly have night-time immersion uptaked more atmospheric $CO_2$ with the tidal influence (Table 4). However, this unpublished result was possible provided that the salt marsh switched from a source to a sink of $CO_2$ during night-time immersion due to water undersaturation with respect to the atmosphere. In a salt marsh on Sapelo Island (USA), Nahrawi et al. (2020) highlighted tidal $CO_2$ flux 625    reductions all year round by distinguishing neap tide and spring tide periods. Their results showed that the highest and lowest reductions in C uptake occurred in spring (-34%) and summer (-13%), respectively, with a similar but greater tidal influence on the C uptake values compared to our study.

To better constrain the tidal influence on the metabolism of the salt marsh, further investigations have been carried out in 2021 in parallel with our EC measurements, with the construction of a digital field model for water heights that can be 630    used to spatially determine, over the whole EC footprint, the exact areas of immersion and emersion (especially for the low water levels) of the marsh in each sector at a 10-min. step. Similarly, during marsh immersion, EC measurements do not directly capture $CO_2$ fluxes from benthic metabolism because of the physical barrier of the water and the lower $CO_2$ diffusion rates in water than in air. Consequently, at the same time as when the NEE measurements were taken, water $pCO_2$



and inorganic and organic carbon concentrations associated with planktonic metabolism were determined each season
through 24-hour cycles to provide essential information on the contribution of planktonic communities and plants to $CO_2$
fluxes during immersion (Mayen et al., in prep.). The lateral C export from salt marshes through tides plays a significant role
in the coastal ocean C cycle (Guo et al., 2009; Wang et al., 2016). Plant respiration and microbial mineralisation of marsh
NPP could generate DIC in water associated with a strong benthos-pelagos coupling. Thus, our 2021 measurements of the
carbon parameters, planktonic metabolism (production/respiration) and other relevant biogeochemical variables over 24-h
diurnal cycles, along with measurements of the soil compartment (root OM production *vs* mineralization; Arnaud et al.,
submitted 2022) carried out simultaneously in the EC footprint would allow for a more integrative calculation of the studied
marsh carbon budget (Mayen et al., in prep.). One advantage of the EC measurements is the aggregation of $CO_2$ fluxes from
all compartments (waterbodies, soil, plants, atmosphere) in salt marshes. Yet, through this flux aggregation, we cannot
mechanistically understand each marsh compartment, and therefore it can be challenging to predict $CO_2$ fluxes under
multiple global changes. Therefore, future contributions should try to simultaneously quantify all these compartments,
especially soil as it is where most of the carbon is stored in salt marshes (Arnaud et al., submitted 2022).

## 5.    Conclusion

In this study, we used the micrometeorological eddy covariance technique to investigate the net ecosystem $CO_2$
exchanges (NEE) at different timescales and to determine the major biophysical drivers of a rewilded tidal salt marsh. Over
the year 2020, the net C uptake from the atmosphere (-483 g C $m^{-2}$ $yr^{-1}$) was mainly related to a low OM decomposition rate
coupled with an intense autotrophic metabolism of halophile plants, especially during the growing season, driven by light,
temperature and VPD. In summer, the brightest days increased the plant GPP and simultaneously, high temperature and VPD
values favoured $R_{eco}$ resulting in a lower net $CO_2$ uptake in summer than in spring. At the daily scale, the tidal rhythm
significantly influenced NEE fluxes according to the level of marsh immersion and PAR. During daytime, tides strongly
limited atmospheric $CO_2$ uptake, up to 90% reductions whereas night-time immersion inhibited atmospheric $CO_2$ emissions
through plant and soil respiration, sometimes even causing a change in metabolic status from source to sink. However, at the
annual scale, NEE flux partitioning into NEP highlighted that the tidal rhythm did not significantly affect the net marsh C
balance. Our continuous NEE measurements have made it possible to better understand the biogeochemical functioning of
salt marshes over a wide range of environmental conditions and have provided essential information on NEE fluxes in
marshes undergoing potential future changes such as global warming or sea level rise.




**Appendix A**

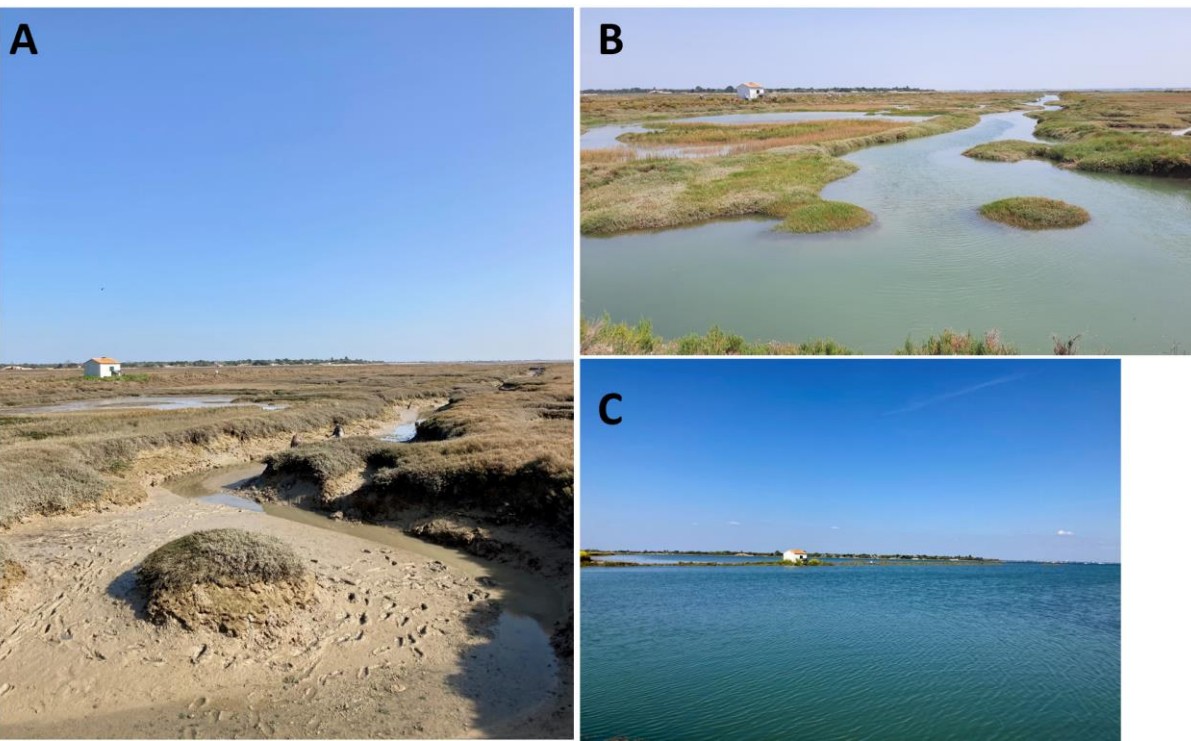

**Fig. A.1. Pictures of the Bossys perdus salt marsh during emersion (A; 01/03/2021 15:00, Hw = 0 m) and immersion (B, 22/07/2021 13:00, Hw = 0.3 m; C, 27/04/2021 16:00, Hw = 1.8 m). Picture A was taken at low tide when all the marsh plants were emerged into the atmosphere. During this time, the channel drains the upstream marsh waters to the estuary. Picture B was taken during incoming tide when advected coastal waters completely fill the channel and immerge the marsh. Picture C was taken at high tide during the highest tidal amplitude when all the marsh plants were immerged by coastal waters. Water heights (Hw) were measured from the STPS sensor located on the salt marsh and not in the channel (see M&M section and Fig. 2). © P. Polsenaere.**










**Table A.1. Estimation of the parameters used for NEE flux partitioning ($a_1$, $a_2$, $R_0$ and $b$) during emersion at the monthly scale. The $a_1$ coefficient is directly linked to the phenology of the ecosystem.**

|  | $a_1$ | $a_2$ | $R_0$ | $b$ |
|---|---|---|---|---|
| January | -7.82 | 370 | 0.34 | 0.04 |
| February | -9.89 | 435 | 0.64 | 0.03 |
| March | -9.38 | 506 | 0.17 | 0.15 |
| April | -12.51 | 787 | 0.24 | 0.12 |
| May | -13.41 | 812 | 0.35 | 0.10 |
| June | -14.68 | 846 | 0.68 | 0.06 |
| July | -14.98 | 934 | 0.84 | 0.05 |
| August | -17.91 | 1397 | 0.56 | 0.07 |
| September | -16.86 | 1419 | 0.32 | 0.09 |
| October | -13.08 | 766 | 0.58 | 0.06 |
| November | -14.37 | 783 | 0.19 | 0.14 |
| December | -7.60 | 360 | 0.31 | 0.09 |





**Table A.2. Monthly mean (µmol $CO_2$ $m^{-2}$ $s^{-1}$) and monthly cumulative (g C $m^{-2}$) fluxes of the measured NEE and estimated NEP during marsh emersion periods (Hw = 0 m).**

| | Mean NEE (µmol $CO_2$ $m^{-2}$ $s^{-1}$) | Mean NEP (µmol $CO_2$ $m^{-2}$ $s^{-1}$) | Cumulative NEE (g C $m^{-2}$) | Cumulative NEP (g C $m^{-2}$) |
|---|---|---|---|---|
| January | -0.75 | -0.77 | -18.2 | -18.7 |
| February | -1.55 | -1.56 | -34.8 | -35.0 |
| March | -1.53 | -1.53 | -37.3 | -37.3 |
| April | -1.95 | -1.93 | -45.2 | -44.9 |
| May | -2.16 | -2.16 | -53.0 | -53.2 |
| June | -2.29 | -2.30 | -50.6 | -50.9 |
| July | -2.34 | -2.33 | -57.7 | -57.6 |
| August | -1.24 | -1.27 | -29.9 | -30.6 |
| September | -1.14 | -1.13 | -27.2 | -26.9 |
| October | -0.80 | -0.80 | -18.4 | -18.4 |
| November | -0.63 | -0.61 | -14.8 | -14.4 |
| December | -0.40 | -0.40 | -6.3 | -6.2 |





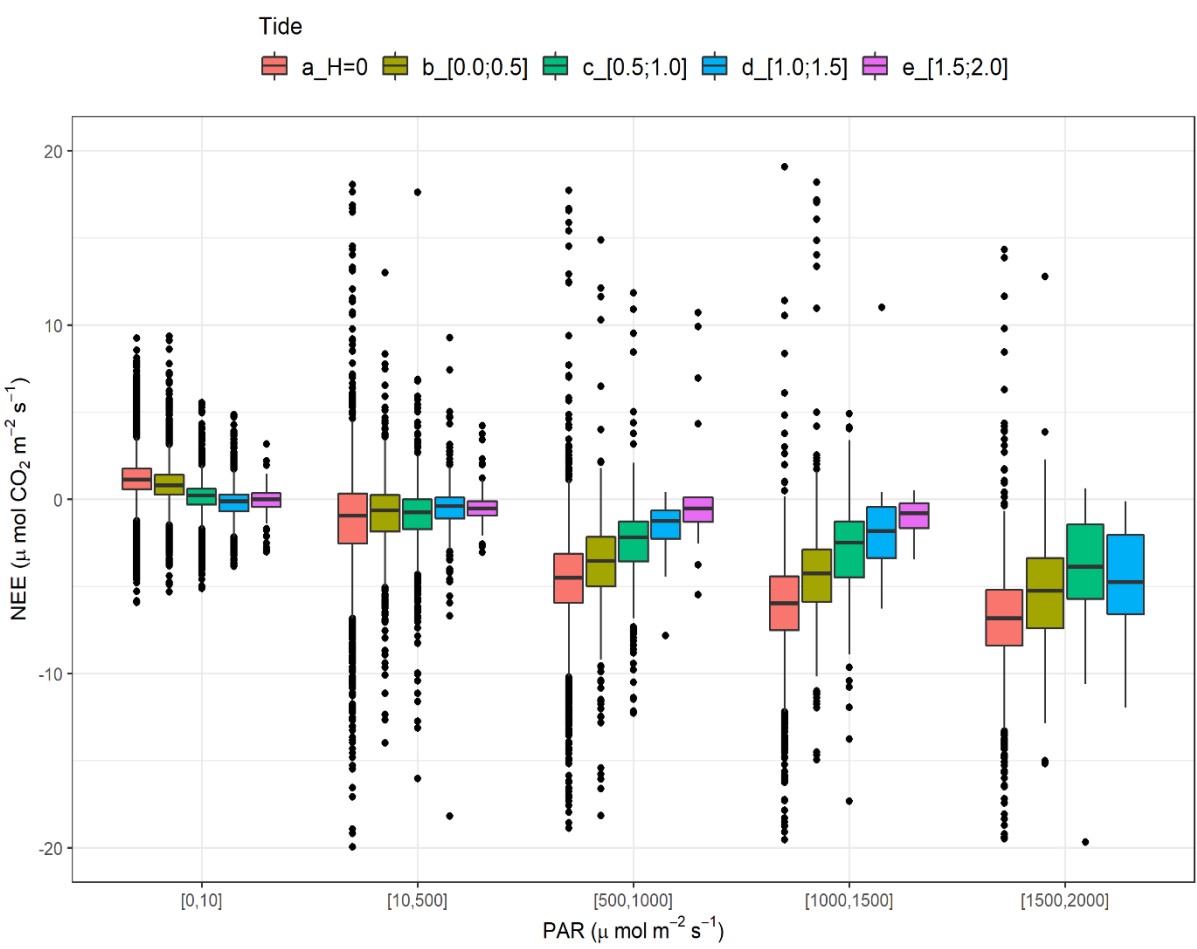

**Fig. A.2. Diurnal/tidal variations (boxplots) of NEE fluxes (µmol $CO_2$ m$^{-2}$ s$^{-1}$) during marsh emersion (Hw = 0 m) and at four water level ranges of 0.5 m within five PAR groups. The five PAR groups are 0 < PAR ≤ 10 (night), 10 < PAR ≤ 500, 500 < PAR ≤ 1000, 1000 < PAR ≤ 1500, 1500 < PAR ≤ 2000 µmol m$^{-2}$ s$^{-1}$.**








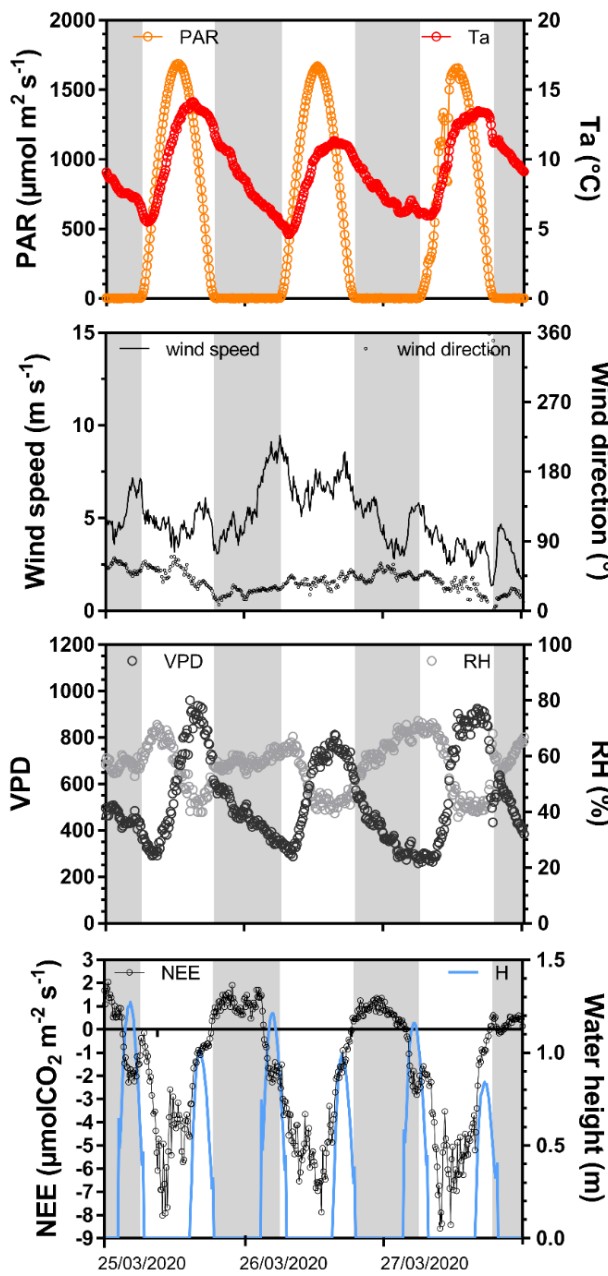

**Fig. A.3. Temporal variations of PAR (µmol m$^{-2}$ s$^{-1}$), Ta (°C), wind speed (m s$^{-1}$), wind direction (°), VPD, RH (%), NEE (µmol CO$_2$ m$^{-2}$ s$^{-1}$) and water height (Hw, m) values measured at a 10-minute frequency in early spring 2020 from 25/03/2020 (00:00 am) to 27/03/2020 (23:50 pm). Grey areas correspond to night-time periods (PAR ≤ 10 µmol m$^{-2}$ s$^{-1}$). This temporal window in March 2020 was chosen to highligh the marsh CO$_2$ sink during night-time immersion in winter, the rapid decrease of CO$_2$ uptake during daytime immersion and the negative correlation between NEE and PAR.**




**Data availability**

All raw data can be provided by the corresponding authors upon request.


**Author contribution**

TLL and PP allowed the funding acquisition. PP, EL and JMB conceptualized and designed the study. JM and PP compiled and prepared the datasets. JM and PK performed statistical and time-series analyses. JM, PP, EL and PK investigated and analysed the data. PK and RC performed the Random forest model. JM, PP, EL, PK, ARG and PS confirmed the data. PP,
EL, MA, JMB, PG, JG and RC provided resources. JM performed the graphics and wrote the manuscript draft. PP, EL, MA, PK, RC, ARG and PS reviewed and edited the manuscript. PP, ARG and PS supervised the PhD thesis of JM.

**Competing interests**

The authors declare that they have no conflict of interest.

**Acknowledgements**

I would like to thank Ifremer (the French research institute for exploitation of the sea) for financing my PhD thesis (2020-
2023). We are grateful to our colleagues who contributed to the fieldwork carried out during this study. This work is a contribution to the Jérémy Mayen's PhD thesis and the ANR-PAMPAS project (Agence Nationale de la Recherche « Evolution de l'identité patrimoniale des marais des Pertuis Charentais en réponse à l'aléa de submersion marine », ANR-18-CE32-0006). The proofreading of the manuscript and the correcting of the English content were carried out by Sara Mullin (PhD; freelance translator).

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
