# Peer review of "Atmospheric CO2 exchanges measured by Eddy Covariance over a temperate salt marsh and influence of environmental controlling factors"

_EGUsphere, 2023_

## Author Comment (AC1)

**RC1**: ['Comment on egusphere-2023-1641'](), Anonymous Referee #1, 07 Aug 2023

Mayen et al present one year of eddy covariance measurement of $CO_2$ exchange in a temperate salt marsh located at the French Atlantic coast. They assess the net $CO_2$ uptake on different time scales (diurnal, tidal, seasonal), analyze environmental controls and particularly focus on the impacts of tides on the atmospheric exchange. They find, compared to published values from other tidal wetlands, a large annual net $CO_2$ uptake.

Coastal wetlands are important for regulating biogeochemical cycling in the coastal zone. C flux studies are still relatively rare, and this study presents a tidal system with different vegetation (and tidal range) than previously published. As such, the study is a good fit for Biogeosciences.

We appreciate Referee#1 comment corroborating our Biogeosciences journal choice to publish this study and fitting the associated manuscript we revised accordingly (see below).

**Major comments:**

C1: The definition of NEP and NEE should be considered carefully in this case (Chapin et al. 2006). Tidal systems tend to export DIC, which is not captured by eddy covariance measurements, so it is better to use NEE instead of NEP.

We thank Referee#1 for this important comment. In terrestrial ecosystems, NEE fluxes from atmospheric eddy covariance (EC) measurements generally correspond to NEP fluxes (Kowalski et al., 2003; Chapin et al., 2006). However, in intertidal ecosystems, like the studied salt marsh, the later relationship is more complex and NEE measured by EC does not fully correspond to NEP. Indeed, water column processes and fluxes such as lateral DIC exports related to marsh heterotrophic respiration, organic matter mineralisation, carbonate dissolution and benthos-pelagos couplings (Wang et al., 2016) are not capture by EC measurements, especially during transient tidal phases (flooding/ebbing). Thus, in our revised manuscript, we replaced NEP by $NEE_{marsh}$. In our study, NEE fluxes correspond to net vertical $CO_2$ exchanges measured by EC whereas, $NEE_{marsh}$ fluxes correspond to net vertical $CO_2$ exchanges estimated at the marsh-atmosphere interface without any tidal influence.

C2: Random Forest Model for gap-filling: It is not clear whether the data set was split in day and night time. The relatively bad night time model performance could be partially due to PAR being 0, if PAR is still included as driver. In addition, it would be helpful if an uncertainty for the resulting annual budget was presented.

In our study, the Random Forest model did not run separately for daytime and night-time data. We used only one model driven on our full 2020 dataset for gap-fill the 18.3% of missing data, with environmental variables easily available or measured, and identified in the literature to significant control $CO_2$ fluxes in salt marshes (RF2 in L259-262; Table A.1). Referee#1 is right, our Random Forest model estimated less well NEE data at night (38%) than at day (59%) that corresponds to only 20% of night-time NEE data. This lower performance model for night-time periods for which one of the most important NEE controlling factors (i.e. light) is absent, may

be explained by other processes in presence of waters, as for instance those occurring in winter during night-time (at both emersion and immersion) related to negatively measured NEE fluxes. These weak night-time $CO_2$ sinks observed at the studied site can be explained by inflow of coastal water undersaturated in $CO_2$ with respect to the atmosphere and/or $CaCO_3$ dissolution in waters or sediments. These aquatic processes and their influence on atmospheric fluxes, now more deeply discussed in the revised sections of the manuscript (see sections below), could participate to the difficulty of the Random Forest model to gap-fill these EC night-time data. To be noticed that at the diurnal scale, machine learning approach for predicting ecosystem $CO_2$ assimilation, even over terrestrial ecosystems in absence of water influence, are in any cases particularly complicated to effectively apply due to non-stationarities coming from multiple processes (see Bartolomeis et al. 2023, https://egusphere.copernicus.org/preprints/2023/egusphere-2023-1826/egusphere-2023-1826.pdf).

According to the uncertainty for the resulting annual budget, we computed a value of 0.42% by comparing the cumulative measured NEE (-462 g C m$^{-2}$) and the corresponding cumulative gap-filled NEE (-453 g C m$^{-2}$) on the 81.5% of our dataset where we had both measured and gap-filled values (i.e. 43264 values on 52704 measurements over the year 2020). The difference between EC and Random Forest model cumulative NEE values (1.95%) was then divided by the 43264 values (mean difference for one measurement) and finally multiplied by 9440 values (difference for 9440 gap-filled values over the whole 2020 dataset) to obtain the 0.42% of uncertainty.

**Section 3.3. Environmental parameters and NEE fluxes at diurnal and tidal scales** (this sentence was added in the revised MS)

p17, L401-L405: "In winter, during some night-time periods, weak $CO_2$ sinks were recorded both during emersion (-0.79 ± 0.84 µmol m$^{-2}$ s$^{-1}$; 137 hours over 71 days) and immersion (-0.82 ± 0.91 µmol m$^{-2}$ s$^{-1}$; 143 hours over 55 days associated with a mean Hw of 0.80 m; Fig. A.2) of the salt marsh. The maximal $CO_2$ uptakes were -4.80 and -5.31 µmol m$^{-2}$ s$^{-1}$ during night-time emersion and night-time immersion, respectively (Table 3)."

**Section 4.2. Metabolism processes and controlling factors at multiple timescales** (this paragraph was added in the revised MS)

p28, L627-L635: "In winter, negative NEE fluxes were measured during some night-time emersion periods in the absence of any photosynthetic processes (18.5% in January, 18.1% in February and 10.7% in March). These negative fluxes could have two mainly sources: (1) an inorganic $CO_2$ diffusion and dissolution processes in saline/alkaline soils over mudflats (Ma et al., 2013) and (2) an inflow of coastal waters undersaturated in $CO_2$ with respect to the atmosphere within the footprint area (in channel for instance; Fig. 2) but not seen by the STPS probe due to our one-location water height measurement and immersion marsh heterogeneity (see 2.2 section). The negative values during night-time emersion could reduce the night-time random Forest model performance for EC data gap-filling and produce an underestimation of

respiration coefficients for NEE flux partitioning (particularly *b*) even causing negative coefficient (February; Table A.2)."

**Section 4.2. Metabolism processes and controlling factors at multiple timescales**

p29, L661-L666: "In our case, night-time $CO_2$ exchanges were reduced up to 100% (completely suppressed), sometimes even causing a change in metabolic status of atmospheric $CO_2$ from source to sink, especially in winter when the $R_{eco}$ rates were the lowest (Fig. 8). The presence of tidal waters advected from the shelf during the night and $CO_2$ undersaturated with respect to the atmosphere due to previous phytoplankton production and/or $CaCO_2$ dissolution in the water column during the day (Gattuso et al., 1999; Polsenaere et al., 2012), could induce a sink which may lead to a net uptake of $CO_2$ at night (Fig. 8)."

C3: Use of RH instead of VPD: The use of RH (%) as driver is not clear to me. Why not use VPD instead? It bypasses the problem of relative scale in RH as well as its temperature dependence.

Referee#1 is right. In the revised manuscript, we replaced RH by VPD, especially for the assessment of environmental drivers on NEE (see revised sections below; Fig. 6). We highlighted a significant correlation between NEE and VPD inducing a decrease of marsh $CO_2$ uptake for highest VPD values (especially, during warm and dry periods). Please refer to the revised sections below.

**Section 3.3. Environmental parameter and NEE flux variations at diurnal and tidal scales** (this sentence was modified in the revised MS)

p16, L382-L385: "Over all seasons, similar diurnal variations in measured NEE and estimated $NEE_{marsh}$ were recorded with, on average, a rapid increase in $CO_2$ uptake during the morning up to the middle of the day (low Ta and VPD values) and then, a decrease in $CO_2$ uptake during the afternoon (high Ta and VPD values) to become a $CO_2$ source during night-time (Figs. 5 and A.2)."

**Section 3.4. Influence of environmental drivers on temporal NEE variations** (this sentence was added in the revised MS)

p18, L426-L429: "During daytime, Vapor Pressure Deficit (VPD) was negatively correlated with NEE (-0.31; n = 27160, p < 0.05) producing a large reduction of $CO_2$ uptake for all PAR levels and even led to a switch from sink to source of atmospheric $CO_2$ from VPD > 1200 Pa for low PAR levels (PAR $\leq$ 500 $\mu$mol m$^{-2}$ s$^{-1}$; Fig. 6-B)."

**Section 4.2. Metabolism processes and controlling factors at multiple timescales** (these paragraphs were modified in the revised MS)

p27, L601-L602: "At the studied salt marsh, we showed a significant influence of VPD and RH on daytime NEE variations favouring plant $CO_2$ uptake for the lowest VPD values (< 1000 Pa) and the highest RH values (> 80%)"

p28, L617-L622: "At our studied site, the highest negative correlations between NEE and PAR were highlighted for low daytime PAR values, indicating that increases in light during the morning strongly favoured $CO_2$ uptake mainly through plant photosynthesis up to the middle of the day. During the afternoon, the high Ta and VPD values (warm and dry periods) produced a reduction of photosynthetic rates through stomatal closure of the C3 plants (Lasslop et al., 2010). This GPP decrease associated with a $R_{eco}$ increase in afternoon reduced the net $CO_2$ uptake up to reach $CO_2$ emissions during night-time (Knox et al., 2018; Xi et al., 2019)."

**Minor comments:**

92: But see published studies such as Kathilankal et al. 2008, Moffett et al. 2010, Forbrich & Giblin 2015, Knox et al. 2018, Nahrawi et al. 2019, Hawman et al. 2023

Yes, please refer to our response to the major comment C1 above. In our revised manuscript, we replaced NEP by $NEE_{marsh}$ since NEE measured by EC do not fully correspond to NEP due to DIC lateral export.

106: Considering that land use impacts are being discussed later in the manuscript, it would be helpful here to get a better idea of the on-going restoration, if that is possible.

The Bossys perdus salt marsh is not under on-going restoration. For several centuries, it was used for salt farming and oyster farming; but since 1981, the salt marsh is protected within the maritime part of the national natural reserve (NNR) to restore the natural site hydrodynamics and marsh halophile vegetation without major restoration work here. The specific typology of the marsh due to past human activities (channel network, humps and dykes) remains and induces an immersion/emersion marsh heterogeneity on the site due to the specific associated microtopography. We modified the revised manuscript (see below) to give readers a better understanding of the history of the site and its current management practice. Moreover, we added pictures of the studied salt marsh during emersion to better visualize the specific typology and halophyle vegetation (Fig. 2).

**Section 2.1. Study site** (this paragraph was modified in the revised MS)

p4, L111-L115: "Between the 17th and 20th centuries, the salt marsh has experienced successive periods of intensive land-use (salt harvesting, oyster farming) and returns to natural conditions before becoming a permanent part of the NNR since 1981 for the biodiversity protection without major marsh restoration work (Gernigon, personal communication). It is

currently managed to restore its natural hydrodynamics while conserving the site's specific typology due to past human activities (channel network, humps and dykes; Fig. 2)."

Is the flux tower site directly or rather indirectly influenced by it? Is the inundation pattern influenced by it and/or vegetation composition?

The Bossys perdus salt marsh did not have a major restoration work (please refer to the responses above). However, past human activities and water management practices for salt farming have shaped the marsh typology (channel network, humps, dykes) and associated microtopography, producing a time-delayed immersion of plants and muds between high and low marsh areas particularly during spring tides. Moreover, we added pictures of the studied salt marsh during emersion periods to better visualize the halophyle vegetation between high marsh levels (*H. portulacoides* and *S. vera*) and low marsh levels (*S. maritima* and mudflats; Fig. 2). We modified the revised manuscript and we discussed more the impact of this immersion/emersion marsh heterogeneity on measured NEE fluxes (see revised sections below).

**Section 2.3. Footprint estimation and immersion/emersion marsh heterogeneity** (this paragraph was modified in the revised MS and the last sentence was added)

p8-9, L207-217: "At incoming tide, when coastal waters begin to fill the channel and then overflow over the marsh (from 0.5 h in spring tides to 2.5 h in neap tides; data not shown), the SSW sector (Fig. 2) was first immersed and a non-zero Hw value was measured. However, although some marsh sectors were immersed at the same time, others were still emerged. Indeed, lowest marsh levels (56% of the footprint area), mainly composed of mudflats and *S. maritima* (Table 1 and Fig. 2), were quickly immersed from Hw > 0 m (south) whereas the whole marsh immersion (muds and overall plants) only occurred 0.75 h later from Hw > 1.0 m at high tide during spring tide. Thus, highest marsh levels (44% of the footprint area), mainly composed of *H. portulacoides* and *S. vera* (Table 1 and Fig. 2), were still emerged for 0 < Hw < 1.0 m. Conversely, at neap tide, this footprint immersion versus emersion marsh heterogeneity could still be present even at high tide due to insufficient water levels. Although, a digital field model for water heights could not be performed in 2020 to have a better spatial representation of the immersion/emersion footprint, all these important considerations were considered in our interpretations in this study."

**Section 4.1. Marsh CO$_2$ uptake and influence of management practices** (this sentence was added in the revised MS)

p25, L436-439: "Thus, due to this emersion/immersion heterogeneity, mud and *S. maritima* areas were quickly immersed by coastal waters whereas, the whole immersion of marsh habitats only occurred during the highest tidal amplitudes favouring a higher atmospheric CO$_2$ uptake by *H. portulacoides* and *S. vera*"

Also, it would be good to describe here the vegetation. It looks like that the two main species are evergreen shrubs, which will be good to highlight here.

In the revised manuscript, we added the description of the marsh vegetation (evergreen/perennial plants) and associated metabolic pathways. Please refer to the revised section below.

**Section 2.1. Study site** (these sentences were added in the revised MS)

P5, L134-139: "The marsh vegetation assemblage was mainly composed by three halophytic species as perennial plants (*Halimione portulacoides*, *Spartina maritima* and *Suaeda vera*; Fig. 2) that associated with different metabolic pathways (the C3-type photosynthesis for *H. portulacoides* and *S. vera* and the C4-type photosynthesis for *S. maritima*; Duarte et al., 2013, 2014). Whereas *H. portulacoides* and *S. vera* are evergreen plants throughout the year, the growing season for *S. maritima* was shorter (from spring) with a flowering period between August and October (plants persist only in the form of rhizomes in winter and fall; Gernigon, personal communication)."

Also, if that information is available, it would be helpful to explain here, what the tide range is and how high/low the marsh surface is located within that tide range.

In the revised manuscript, the maximal tidal range of the Fier d'Ars estuary (5 meters) was added (see revised section below).

**Section 2.1. Study site** (this sentence was modified in the revised MS)

p4, L115-116: "This salt marsh is linked to the Fier d'Ars tidal estuary that exchanges between 2.4 and 10.2 million $m^3$ of coastal waters with the Breton Sound continental shelf allowing a maximal tidal range of 5 m in the estuary (Bel Hassen, 2001)."

I assume that 0m as reported in the manuscript refer to the marsh surface?

Referee#1 is right, the water height (Hw) values reported in the manuscript were measured by the STPS probe (SSW wind sector; Fig. 2) and referred to the marsh surface.

144-160: This chapter is probably not really necessary, or can be briefly summarized in section 2.3

In the revised manuscript, we reduced the chapter on the Eddy Covariance theory which is well-established in the literature (Aubinet et al., 1999; Baldocchi, 2003; Aubinet et al., 2012; Burba,

2021) and we summarised this theory part in the beginning of the section 2.2. Eddy Covariance and micrometeorological measurements (see revised section below). However, we chose to conserve the theory equation of flux calculations method as well as the associated assumptions to ensure that the Eddy Covariance's technique is well understood by a large number of readers. We removed lines 144-148 and lines 145-160 from submitted manuscript (please refer to Referee#2 comment responses).

**Section 2.2. Eddy Covariance and micrometeorological measurements** (this paragraph was modified in the revised MS)

p7, L153-155: "The atmospheric eddy covariance (EC) technique allow to quantify the net $CO_2$ fluxes at the ecosystem-atmosphere interface through micrometeorological measurements of the vertical component of atmospheric turbulent eddies (Aubinet et al., 1999; Baldocchi, 2003; Burba, 2021)."

186-190: How are the periods of flooding treated in the footprint model? Since the flood height seems to be substantial, a constant measurement height seems doubtful. Is the flooding filtered out?

We thank Referee#1 for her/his important comment. In our study, we have chosen to use a constant measurement height (Zm) for the footprint estimation (Zm = 3.15 m; L196) because the Bossys perdus salt marsh is in majority terrestrial (74.5% of the time over the year 2020; Table 2) and during most high tide periods, coastal waters are mainly located in channels representing a minor immersion area (below 7%; Table 1) without influencing Zm. Indeed, marsh immersion could influence Zm only for the highest tidal amplitudes when Hw > 1.5 m (less than 1% of the time over the year 2020; Table 2), thus this time period can be negligible. Moreover, given the accuracy of the Hw measurements (± 0.3 m; L186 in the revised MS), we did not wish to add any further uncertainty to the footprint estimate. For this reason, we found it consistent to use a constant measurement height (Zm = 3.15 m) rather than using a variable Zm taking into account water height values measured by the STPS probe or using only data at emersion (Hw = 0 m) for footprint calculation. For comparison and verification, we performed these two footprint estimations both with variable Zm (using Hw measurements) and constant Zm (using data at emersion) and we obtained exactly the same footprint shape and extend as the one in the submitted manuscript. Thus, we have chosen to conserve the footprint from the submitted MS.

**Section 2.3. Footprint estimation and immersion/emersion marsh heterogeneity** (this sentence was added in the revised MS)

p8, L197-198: "For verification, we performed the footprint estimations both with variable Zm from water height measurements and with constant Zm from data at emersion and we obtained the same footprint shapes and extends."

Also, how does the displacement height relate to vegetation/canopy height, based on Fig. A1-A3, the canopy height is very short?

Referee#1 is right, the canopy height of predominant plants on the studied salt marsh is short (i.e. 0.15 m for *Halimione portulacoides* and 0.30 m at maximum for *Spartina maritima*) as you can see in figure A.1. In our study, instead of using a displacement height (d) value hardly found in the literature for a comparable ecosystem, d was calculated by the EddyPro software from 0.67 times the canopy height according to the following equation (LI-COR *Inc.*):

$d = 0.67 \times$ canopy height $= 0.67 \times 0.15 = 0.10$ m.

As halophile plants (*Spartina maritima*, *Halimione portulacoides* and *Suaeda vera*) at the studied site had a low phenological variation at the monthly scale (Table A.2), we did not record significant variation of canopy height neither over the year, contrarily to the *Spartina alterniflora* specie in some U.S. salt marshes (Nahrawi et al., 2020). Thus, in our study, the displacement height was set constant (d = 0.10 m; L194 in the revised MS). In the revised manuscript, we added the canopy height in the caption of the figure 2 (L145 in the revised MS) and the calculation of the displacement height estimation in the footprint estimation section (see revised section below).

**Section 2.3. Footprint estimation and immersion/emersion marsh heterogeneity**

p8, L192-197: "Footprints were estimated using the model of Kljun et al. (2015) applied to data from the year 2020 to obtain an annual averaged footprint from the constant measurement height ($Z_m$, 3.15 m), the constant displacement height (d = 0.1 m, estimated from 0.67 times the canopy height; LI-COR *Inc.*), mean wind velocities (u_mean, m s$^{-1}$), standard deviations of the lateral velocity fluctuations after rotation [sigma_v, m s$^{-1}$], the Obukhov length (L), friction velocities (u$^*$, m s$^{-1}$) and wind directions (°) obtained from the EC measurements and the EddyPro processing software (EddyPro® v7.0.8, LI-COR Inc.) output."

212: I understand the reasoning for the shorter than usual averaging period, but I would also like to see a statement on how much that impacts the frequency correction, maybe add an average FC correction factor.

Indeed, as Referee#1 understood, we have chosen a time average of 10 minutes due to strong fluctuations of high-frequency EC data during periods of incoming and ebbing tides. EC data were calculated with the EddyPro software and a correction for flux spectral losses in the low frequency range was performed according to Moncrieff et al. (2004). This precision concerning EC data correction in the low frequency range was added in the revised manuscript in p9, L234-235 (see revised section below).

**Section 2.4. EC data processing and quality control** (this sentence was added in the revised MS)

p9, L235-236: "During the EC data processing by EddyPro, a correction for flux spectral losses in the low frequency range was performed according to Moncrieff et al. (2004)."

Moncrieff, J. B., R. Clement, J. Finnigan, and T. Meyers. 2004. Averaging, detrending and filtering of eddy covariance time series, in Handbook of micrometeorology: a guide for surface flux measurements, eds. Lee, X., W. J. Massman and B. E. Law. Dordrecht: Kluwer Academic, 7-31.

Also, over two time periods without tidal immersion (neap tides) associated with strong variations in wind speeds, we compared NEE fluxes computed for an averaging time of 10 mins *versus* 30 mins. We calculated the ratio "average of the 3 values at 10 mins / the value at 30 mins" and we recorded that the ratio is close to 1 for highest wind speeds ($1.07 \pm 0.88$ for $5 <$ wind speeds $< 10$ m s$^{-1}$) and remains at 1 for lowest wind speeds ($1.01 \pm 0.13$ for $0 <$ wind speeds $< 5$ m s$^{-1}$). Thus, we can assume that there has been no significant loss at low frequencies in our 10 min. averaged fluxes using the Moncrieff et al. 2004 correction in EC data processing.

Other authors had also chosen a time average of 10 mins like Polsenaere et al. (2012) in the tidal bay of Arcachon where they showed no significant loss of low frequencies neither using this time averaging in these highly variable tidal environments.

233: Is the Random Forest model run/fitted separately for day and nighttime data?

In our study, the Random Forest model did not run separately for daytime and night-time data. There is only one model for the gap-filling of all EC data, with environmental predictors identified in the literature to control $CO_2$ fluxes in salt marshes (see response to the major comment C2 above).

239-240: Please give the results of this analysis and how you define 'sufficient'.

In fact, the word 'sufficient' is not well adapted here; we modified the submitted manuscript to better understand the choose of the gap-filling model in this study (L266-267). We used the Random Forest 2 (RF2) model with PAR, Ta, Hw and RH as environmental predictors because its performance indicators showed a high Pearson coefficient ($R^2 = 0.88$) and low values of root mean square error (RMSE = 1.27) and model bias (0.0024) allowing to correctly gap-fill a large EC data (Table A.1). The RF3 model had the highest $R^2$ and the lowest RMSE but we didn't have continuous data of wind direction throughout the year for this using this model.

**Section 2.5. Flux gap-filling and statistic tools** (this section was modified in the revised MS)

p10, L251-271: "The random Forest (RF) model was used to gap-fill our EC dataset. Random forest is a supervised machine learning technique proposed by Breiman (2001) that can model a non-linear relationship with no assumption about the underlying distribution of the data population. This method has been shown to be particularly suited to gap-fill EC data (Kim et al., 2020; Cui et al., 2021). Random forest builds multiple decision trees, each of which are based on a bootstrap aggregated data sample (i.e. bagging of the EC data) and a random subset of predictors (i.e. the selected environmental data; Table A.1). We build random forest models with environmental predictors that have been identified in the literature to control $CO_2$ fluxes in salt marshes and which were available during the gaps and with measurements recorded between 2019 and 2020 (Table A.1). Each random forest model was built from a trained bagging ensemble of 400 randomly generated decision trees (Kim et al., 2020) with the "randomForest" package in the R software (Liaw and Wiener, 2002). In this study, we used the RF2 model with PAR, air temperature, water height and relative humidity as environmental predictors because its performance indicators showed a high Pearson coefficient ($R^2$ = 0.88) and low values of root mean square error (RMSE = 1.27) and model bias (0.0024) allowing to correctly gap-fill a large EC data (Table A.1). The calculated uncertainty of the RF2 model on the resulting annual C budget was 0.43%. Each tree was trained from bagged samples including 70% of the initial dataset. The remaining 30% of the data were used to estimate the fit of each random forest model. The model used was then able to explain 88% of the variability in the test data. Daytime data were better explained than night-time ones (59% *vs.* 38%), with light being the main parameter of the model. However, only 20% of the night-time EC data were gap-filled with the Random Forest model. Using a partial dependence analysis and an ondelette analysis, we concluded that the relationships and temporal dynamics modelled allowed to correctly fill the gaps in our dataset. However, extreme values of some predictors (i.e. PAR > 1000 µmol m$^{-2}$ s$^{-1}$) can reduce the random Forest model performance for estimation of EC data. This observation is common for random forest models, as they show poor results for extreme values. Other models such as artificial neural networks were also tested but showed poorer results (Table A.1)."

**Table A.1. Performance indicators for each model (RF: Random Forest, ANN: Artificial Neural Network) tested to gap fill the $CO_2$ fluxes. Predictor variables are PAR (Photosynthetically active radiation, µmol m$^{-2}$ s$^{-1}$), Ta (air temperature, °C), Hw (water height, m), RH (Relative Humidity, %) and Vd (wind direction, m s$^{-1}$). The performance indicators are the coefficient of linear determination Pearson which shows the level of variability captured by the model ($R^2$), the racine of the error quadratic average which gives an overview of the uncertainty of the result (RMSE: Root Mean Square Error), as well as the bias of the model.**

| Models | Predictor variable | RMSE | Bias | $R^2$ |
|--------|-------------------|------|------|-------|
| RF1 | PAR, Ta, Hw | 1.42 | 0.0039 | 0.85 |
| RF2 | PAR, Ta, Hw, RH | 1.27 | 0.0024 | 0.88 |
| RF3 | PAR, Ta, Hw, Vd | 1.19 | 0.0029 | 0.90 |
| ANN1 | PAR, Ta, Hw | 1.95 | -0.0003 | 0.71 |
| ANN2 | PAR, Ta, Hw, RH | 1.89 | 0.0021 | 0.73 |
| ANN3 | PAR, Ta, Hw, Vd | 1.81 | 0.0041 | 0.75 |

241: Which variables were less influential at high PAR levels?

In gap-filling models, data resulting from the random forest models are generally less well estimated for predictor extreme values (i.e. PAR) because the model is only good within its own limits (we modified the revised MS; see revised section below). For instance, the model underestimated the impact of the immersion (Hw > 1 m) for the highest PAR levels, where the $CO_2$ sink was up to 90% higher than *in situ* data.

243: If you have run ANN as well, please give the results, maybe add in an appendix to see/justify the use of the random forest model.

In the revised manuscript, we added a supplementary table (Table A.1.; see responses above) showing the performance indicators of the different models (ANN and RF) tested in this study to gap fill NEE fluxes. According to values of Pearson coefficient ($R^2$), root mean square error (RMSE) and the model bias, we have chosen the RF2 model to gap-fill our EC data (18.3% of the EC data were removed and gap-filled with RF2 to obtain continuous flux data in 2020). Please refer to our responses above and the modified section.

247: Why analyze monthly mean values? Please explain the rationale for this analysis.

In our study, we have chosen to perform a pairwise Spearman's correlation analysis on 10 min. values and monthly mean values to assess the influence of the environmental drivers on NEE fluxes at different temporal scales (Schafer et al., 2019). For instance, at the monthly scale (monthly mean values), PAR and Ta displayed the strongest negative correlations with NEE showing that the highest $CO_2$ uptakes were recorded during warm and bright months of the year.

**Section 2.5. Flux gap-filling and statistic tools** (this sentence was modified in the revised MS)

p10, L274-275: "To assess the influence of meteorological and hydrological drivers on NEE at different temporal scales, we performed a pairwise Spearman's correlation analysis on the 10-min. values and monthly mean values ("cor function" in R)."

254-255: Why group into these five PAR classes? Please explain the rationale for this analysis.

In our study, NEE fluxes were grouped into five PAR groups to reduce NEE fluctuations due to PAR variations and to better analyse the potential influence of other environmental drivers on NEE (figures 6 and 7). We modified our revised manuscript in L283-285 for a better explanation of this analysis (see below).

**Section 2.6. Temporal analysis of NEE fluxes and partitioning** (this sentence was modified in the revised MS)

p11, L284-286: "For the NEE flux analysis according to environmental drivers, NEE fluxes were grouped into five PAR groups ($0 < PAR \leq 10$, $10 < PAR \leq 500$, $500 < PAR \leq 1000$, $1000 < PAR \leq 1500$ and $1500 < PAR \leq 2000$ µmol m$^{-2}$ s$^{-1}$) to reduce NEE fluctuations due to PAR variations."

255-267: I am not sure why this description shows up in this chapter. An explanation of tide height and reference point (datum) would be helpful in the site description. The aspect of the spatial heterogeneity (in elevation and distance to channels) probably merits some explanation in the site description or in the footprint analysis section as well.

In the revised manuscript, we followed the advice of the Referee#1 and we moved the explanation of our one-location water height measurements and the associated immersion/emersion marsh heterogeneity in the footprint analysis section (see 2.3. from L205 to L216).

Overall, the last sentence, that this heterogeneity was taken into account merits more detail. How was this done, if there is no digital elevation model?

We thank Referee#1 for her/his important comment. Indeed, the immersion/emersion marsh heterogeneity could not be considered in our calculations and analyses because we could not perform a digital elevation model for the 2020 data. However, this important consideration has been taken into account in our interpretations. This sentence was corrected in the revised MS (see revised section below). In the submitted discussion, we indicated that further works were performed in 2021 EC data to obtain a digital field model for water heights and determine the exact areas of immersion and emersion of the marsh in each sector (L628-631 in the submitted MS).

**Section 2.3. Footprint estimation and immersion/emersion marsh heterogeneity** (this sentence was modified in the revised MS)

p8-9, L215-217: "Although, a digital field model for water heights could not be performed in 2020 to have a better spatial representation of the immersion/emersion footprint, all these important considerations were considered in our interpretations in this study."

270: See above: NEP by definition (Chapin et al. 2006) includes inorganic carbon exchange, which in this system is incompletely resolved (due to DIX exchange/export).

Yes, we agree with Referee#1, please see response to the major comment C1 above. Our estimated NEE fluxes at the marsh-atmosphere interface (NEE flux partitioning from Kowalski

et al. 2003) did no correspond to NEP values due to tidal effects and water column processes that are not taking into account with atmospheric EC measurements. Thus, in the revised manuscript, estimated NEE fluxes at the marsh-atmosphere interface without tidal influence correspond to $NEE_{marsh}$ rather than NEP.

273: Agreed, but there is likely slower porewater DIC transport during marsh exposure as well.

Referee#1 is right, there is likely slower porewater DIC transport during marsh immersion. In 2021, we performed DIC measurements during continuous 24-hour cycles at each season simultaneously with NEE and water $pCO_2$ measurements to precisely answer this question (L631-642 in the submitted MS).

276-285: How is tidal inundation dealt with in this analysis? Are data points during submergence filtered out?

During the NEE flux partitioning, estimations of the GPP coefficients ($a_1$ and $a_2$; Eq. 2) and $R_{eco}$ coefficients ($R_0$ and $b$; Eq. 3) were computed at the monthly scale during emersion without tidal water where measured NEE fluxes correspond to marsh metabolic fluxes ($NEE_{marsh}$). Then, when all parameters used for NEE flux partitioning ($a_1$, $a_2$, $R_0$ and $b$) were estimated for each month, they were used to estimated $NEE_{marsh}$ for each Ta and PAR values measured at a frequency of 10 minutes to obtained $NEE_{marsh}$ signal over the year corresponding to marsh metabolic fluxes without any tidal water (see paragraph in the revised manuscript from L303 to L313).

p11-12, L305-315: "During NEE flux partitioning, estimations of the GPP coefficients ($a_1$ and $a_2$; Eq. 2) and $R_{eco}$ coefficients ($R_0$ and $b$; Eq. 3) were performed by the least square method ("minpack.lm" package in R) at the monthly scale only during emersion periods where measured NEE fluxes correspond to estimated $NEE_{marsh}$ fluxes. Firstly, for each month, $R_0$ and $b$ were estimated during night-time emersion periods where NEE = $R_{eco}$ following Eq. 3 (Wei et al., 2020b). Then, $a_1$ and $a_2$ were estimated during daytime emersion periods using night-time respiration coefficients ($R_0$ and $b$) where NEE = GPP - $R_{eco}$ following Eq. 2 and Eq. 3 (Kowalski et al., 2003). Finally, $NEE_{marsh}$ (marsh metabolic fluxes without tidal influence) were calculated for each PAR and Ta value measured at a 10-min. frequency throughout the year using the monthly coefficients calculated for the partitioning (Eq. 2). As our ecosystem had a low phenological variation (Table A.2), we concluded that a monthly time step for the coefficient estimation was sufficient to answer our study objectives. During emersion periods, monthly net C balances (i.e. budgets) of measured NEE and estimated $NEE_{marsh}$ were very similar as well as the monthly mean fluxes (Table A.3), confirming the correct NEE flux partitioning calculations done in this study."

However, in winter, negative NEE fluxes measured during night-time emersion periods due to inflow of coastal waters undersaturated in $CO_2$ in channels and/or $CaCO_3$ dissolution in marsh sediments could produce an underestimation of respiration parameters for the NEE partitioning (particularly $b$) even causing negative parameter (see revised section below).

**Section 4.2. Metabolism processes and controlling factors at multiple timescales** (this paragraph was added in the revised MS)

p28, L627-L635: "In winter, negative NEE fluxes were measured during some night-time emersion periods in the absence of any photosynthetic processes (18.5% in January, 18.1% in February and 10.7% in March). These negative fluxes could have two mainly sources: (1) an inorganic $CO_2$ diffusion and dissolution processes in saline/alkaline soils over mudflats (Ma et al., 2013) and (2) an inflow of coastal waters undersaturated in $CO_2$ with respect to the atmosphere within the footprint area (in channel for instance; Fig. 2) but not seen by the STPS probe due to our one-location water height measurement and immersion marsh heterogeneity (see 2.2 section). The negative values during night-time emersion could reduce the night-time random Forest model performance for EC data gap-filling and produce an underestimation of respiration coefficients for NEE flux partitioning (particularly *b*) even causing negative coefficient (February; Table A.2)."

323-325: How does this flooding relate to marsh surface and canopy height?

Due to heterogeneous typology and microtopography of our study site (see responses above) associated with meteorological conditions, during high tide periods for instance, some wind sectors can be immersed while other sectors are still emerged at the same time, despite a same canopy height ($d = 0.10$ m). Moreover, due to the incertitude on Hw measurements ($\pm 0.30$ m), it will be difficult to answer this question even with a digital field model for water levels.

478-495: I don't see the need for this chapter. It seems to re-iterate information from the introduction.

In our revised manuscript, we followed the advice of the Referee#1 and we strongly reduced the section 4.1. of the submitted MS to avoid repetitions with the introduction and M&M sections regarding methodological aspects of the Eddy Covariance technique. Thus, we removed lines 484-495 from submitted MS.

511: This is really interesting, but should be mentioned and discussed/analyzed earlier than here.

In the revised manuscript, the time-delayed immersion of plants and muds between high and low marsh areas due to specific marsh typology were more discussed at the beginning of the discussion (L527-534 in the revised MS). This emersion/immersion heterogeneity allows a lower immersion duration of marsh plants during neap tide periods (low tidal amplitudes) favouring a higher marsh $CO_2$ uptake (see the sentence added in the revised MS below).

**Section 4.1. Marsh CO$_2$ uptake and influence of management practices** (this sentence was added in the revised MS)

p25, L536-539: "Thus, due to this emersion/immersion heterogeneity, mud and *S. maritima* areas were quickly immersed by coastal waters whereas, the whole immersion of marsh habitats only occurred during the highest tidal amplitudes favouring a higher atmospheric CO$_2$ uptake by *H. portulacoides* and *S. vera*."

515-524: This is important background information. If NPP (based on literature values) is very close to NEE, no heterotrophic respiration occurs, which seems unlikely. It is a common problem in these systems, that the range in different component fluxes is difficult to constrain.

We thank Referee#1 for her/his important comment. The estimated NPP of *H. portulacoides* and *S. maritima* in Sousa et al. (2010) showed a significant contribution of these halophytic species in our marsh CO$_2$ uptake. However, our net C balance at the annual scale could also be influenced by the heterotrophic respiration of the marsh producing atmospheric CO$_2$ emissions and lateral DIC exports in tidal waters (see the revised sentence below).

**Section 4.1. Marsh CO$_2$ uptake and influence of management practices** (this sentence was completed in the revised MS)

p25-26, L544-547: "Thus, the net metabolism of these halophytic plants could play an important role in our net C balance but, according to "the marsh CO$_2$ pump" (Wang et al., 2016), a significant proportion of marsh NPP was respired by heterotrophic processes in benthos and then (1) emitted as atmospheric CO$_2$ (38 ± 11%) and (2) exported by tides as DIC (37 ± 15%; Song et al., 2023)."

Additional influences from benthos is a possible factor here, but is it large enough? Also, considering the depth of inundation and the length of the submergence, what about lateral exchange? I don't expect that you quantify that, but at least discuss this as possible factor as well.

Referee#1 is right, there is a possible benthos influence along with lateral DIC exports from heterotrophic respiration. The low water level during marsh immersion could favour benthic-pelagic coupling and influence water pCO$_2$ dynamics (see the sentence added in the revised MS above). In 2021, we performed DIC measurements during continuous 24-hour cycles at each season simultaneously with NEE and water pCO$_2$ measurements to precisely answer this question (L631-642 in the submitted MS).

528-539: Here is a place to discuss the fact that there is evergreen vegetation (in contrast to some of the published studies) as well as the question of vegetation density/LAI. From the pictures, it looks like that LAI is not very high, which could be a factor in low Reco values (and NEE).

In the revised manuscript (see the revised paragraph below), we discussed that the halophyte vegetation of the salt marsh, mainly composed of evergreen species, was net autotrophic throughout the year allowing a net $CO_2$ uptake during both the growing and non-growing seasons whereas, the smooth cordgrasses as *S. alterniflora* in some tidal salt marshes was net heterotrophic during winter and fall due to the winter plant senescence, producing in turn, atmospheric $CO_2$ emissions during these periods of the year.

**Section 4.2. Metabolism processes and controlling factors at multiple timescales** (theses sentences were modified in the revised MS)

p26-27, L574-581**:** "In our studied marsh, the halophyle vegetation, mostly composed of evergreen species, was autotrophic throughout the year allowing a net C uptake from the atmosphere during both the growing and non-growing seasons (between 9 g C m$^{-2}$ in December and 73 g C m$^{-2}$ in July) whereas, the senescence of smooth cordgrass plants in some salt marshes (*S. alterniflora* and *S. cynosuroides* for instance) from October produced a marsh heterotrophy and a net C source to the atmosphere in winter and fall (Schafer et al., 2014; Artigas et al. 2015; Forbrich et al., 2018). In our case, *S. maritima* is a perennial specie but with a relatively short growing period, thus during winter and fall, the metabolism of this halophytic plant could have a significant lower influence on marsh C uptake than *H. portulacoides* and *S. vera*."

569-571: Does that mean you can exclude the possibility of stomatal closure in the afternoon (high Ta and high VPD)? This is something that C3 vegetation can be sensitive to (see Lasslop et al. 2010).

No, we can't exclude this possibility; indeed, the high Ta and VPD values measured in the afternoon (warm and dry periods) produced a reduction in photosynthesis of marsh plants, particularly the C3 plants (*H. portulacoides* and *S. vera*), through the closure of their stomata (Lasslop et al., 2010). This GPP decrease associated with a R$_{eco}$ increase in the afternoon reduced the net $CO_2$ uptake by the marsh (Figs. 5 and A.2). Lasslop et al. (2010) showed that high VPD values in the afternoon on warm and dry days lead to a limitation of GPP by stomatal closure and thus to higher carbon uptake in the morning rather than in afternoon. In the revised manuscript, the discussion was completed in L610-613 (see revised paragraph below).

**Section 4.2. Metabolism processes and controlling factors at multiple timescales** (theses sentences were modified in the revised MS)

p28, L619-622: "During the afternoon, the high Ta and VPD values (warm and dry periods) produced a reduction of photosynthetic rates through stomatal closure of the C3 plants (Lasslop

et al., 2010). This GPP decrease associated with a $R_{eco}$ increase in afternoon reduced the net $CO_2$ uptake up to reach $CO_2$ emissions during night-time (Knox et al., 2018; Xi et al., 2019)"

589-593: This is not the result from the current analysis, but offers a way of elaborating the difference in exchange between the water column and the vegetated marsh.

Referee#1 is right. During some daytime immersion periods, our observations showed a temporary increase of marsh $CO_2$ uptake, especially for low immersion levels, but further works are needed to study the contribution of planktonic and benthic metabolisms on these $CO_2$ uptakes during marsh immersion (see the revised paragraphs below).

**Section 4.2. Metabolism processes and controlling factors at multiple timescales** (this paragraph was modified in the revised MS)

p29, L648-652: "Indeed, during the brightest periods in winter and spring, the temporary increases in $CO_2$ uptake recorded during incoming tides could be related to (1) an increase in the GPP of *H. portulacoides* and *S. vera* (highest marsh levels) favoured by VPD and Ta decreases due to tidal conditions and/or (2) tidal waters advected from the shelf that are undersaturated in $CO_2$ with respect to the atmosphere due to phytoplankton blooms (Mayen et al., in prep.)."

**Section 4.3. Salt marsh carbon budgets for future research perspectives** (paragraph from submitted MS)

p30-31, L691-702: "Similarly, during marsh immersion, EC measurements do not directly capture $CO_2$ fluxes from benthic metabolism because of the physical barrier of the water and the lower $CO_2$ diffusion rates in water than in air. Consequently, at the same time as when the NEE measurements were taken, water $pCO_2$ and inorganic and organic carbon concentrations associated with planktonic metabolism were determined each season through 24-hour cycles to provide essential information on the contribution of planktonic communities and plants to $CO_2$ fluxes during immersion (Mayen et al., in prep.). The lateral C export from salt marshes through tides plays a significant role in the coastal ocean C cycle (Guo et al., 2009; Wang et al., 2016). Plant respiration and microbial mineralisation of marsh NPP could generate DIC in water associated with a strong benthos-pelagos coupling. Thus, our 2021 measurements of the carbon parameters, planktonic metabolism (production/respiration) and other relevant biogeochemical variables over 24-h diurnal cycles, along with measurements of the soil compartment (root OM production *vs* mineralization; Arnaud et al., submitted 2022) carried out simultaneously in the EC footprint would allow for a more integrative calculation of the studied marsh carbon budget (Mayen et al., in prep.)."

---

## Author Comment (AC2)

**RC2**: 'Comment on egusphere-2023-1641', Francisco Artigas, 16 Aug 2023

**C1:** Does the paper address relevant scientific questions within the scope of the journal?

The content fits under either Atmospheric Sciences, Biogeosciences, or Climate. It addresses salt marsh ecosystem vulnerability to rising sea levels via assessing carbon sink potential in a temperate salt marsh. It also addresses relevant questions regarding coastal wetlands, the carbon cycle, and the effect of micrometeorological parameters in net ecosystem exchange and net primary production of coastal wetlands.

We are grateful to Referee#2 for his positive feedback. Referee#2 highlighted the importance of our work concerning the assessment of the atmospheric $CO_2$ uptake capacity of salt marshes under changing environmental conditions and during tidal immersion. In blue carbon systems like salt marshes (coastal wetlands vulnerable to climate changes), it is needed to accurately measured net ecosystem $CO_2$ exchanges through long-term $CO_2$ flux timeseries to better understand the influence of biotic and abiotic controlling factors, especially water height levels, in order to include these coastal systems in global carbon budgets and predict future marsh carbon sinks.

**C2:** Does the paper present novel concepts, ideas, tools, or data?

It is novel in that it looks at inundation length as a potential indicator for future carbon sink loss due to rising sea levels. It is also interesting that the presence of succulent evergreens can maintain the carbon sink profile over the winter when most of the tidal marshes turn into sources. Using random forest models to fill gaps in the CO2 flux data is also a relatively new concept.

We are grateful to Referee#2 for his positive feedbacks on our manuscript. Referee#2 is right, although many studies have yet used the eddy covariance method to assess NEE fluxes over salt marshes, only a limited number have quantified the tidal effects on marsh $CO_2$ uptake. Moreover, the utilization of a random forest model to gap fill removed data is relatively new though adapted with atmospheric Eddy Covariance (EC) data (Bartolomeis et al., 2023; https://egusphere.copernicus.org/preprints/2023/egusphere-2023-1826/egusphere-2023-1826.pdf). Using an atmospheric EC system to measured continuously NEE fluxes, we highlighted an annual carbon sink mainly due to photosynthesis of the productive evergreen plants. Our study also provides relevant information on NEE fluxes during marsh immersion by decreasing daytime $CO_2$ uptake and night-time $CO_2$ emissions at the daily scale whereas the immersion did not affect the annual marsh C balance.

**C3:** Are the scientific methods and assumptions valid and clearly outlined?

The authors did a thorough literature review. The introduction is logically constructed, and the study's objectives are clear.

We are grateful to Referee#2 for his positive feedbacks on the introduction and literature review of our manuscript.

The study design raises some questions as there is a high percent cover of mud in the sampled area. The biofilm cover on these mud flats can occasionally act as sinks and are more susceptible to tidal fluctuation than vascular plants regarding carbon fixation. The mud effect and vegetation assemblage pattern should have more discussion. Is the study site representative of the Bossys perdus marsh? Would an area with significantly less mud flat cover give similar results?

The study site is representative of the Bossys perdus salt marsh characterized by a particularly complex assemblage of halophytic plants (*Halimione portulacoides*, *Spartina maritima* and *Suaeda vera*), mudflat areas and secondary channel networks as described in the submitted MS (p11, L299-305, Table 1) due to the site history in particular (see Referee#1 comment responses).

Referee#2 comment with regards to potential vascular plants *versus* mud areas relative contributions to EC measured $CO_2$ fluxes is entirely right and was already on our mind and reflexion. In the revised manuscript, we have more specifically studied these habitat effects (vascular plants *vs.* microphytobenthos) on NEE fluxes, through a new figure (Fig. 7) assessing spatial NEE variations according to each wind sectors, during emersion periods only and during daytime and night-time (see revised sections below). During emersion, we showed a low biofilm metabolism on muds (production and respiration) with lower daytime $CO_2$ uptake and lower night-time $CO_2$ emission than respective $CO_2$ fluxes coming from halophyle plants (Table 1 and Fig. 7). Moreover, in the revised manuscript, we discussed the mudflat's role in carbon fixation/emission under tidal influence and more generally, on the contribution of microphytobenthos on the coastal carbon cycle (see revised sections below).

**Section 3.4. Influence of environmental drivers on temporal NEE variations** (this paragraph was added in the revised MS)

p18, L435-447: "For wind directions, a spatial heterogeneity of NEE was recorded according to wind sectors both during daytime and night-time (Fig. 6-F). Within the footprint area composed of an assemblage of plants and muds (Fig. 2), the highest $CO_2$ uptakes were generally recorded from the Southern sectors (high vegetation:mud ratios) whereas, the lowest $CO_2$ uptakes were generally recorded from the Northern sectors (low vegetation:mud ratios; Fig. 7). For instance, our sectorial NEE analysis during daytime emersion showed that SSE sector (vegetation:mud ratio of 2.4; Table 1) uptaked 32% (winter), 25% (spring) and 50% (fall) times more atmospheric $CO_2$ than NNW sector (vegetation:mud ratio of 0.8; Table 1). Moreover, in winter and fall, we highlighted that $CO_2$ uptake rates of *H. portulacoides* (C3 specie) were significantly higher than *S. maritima* (C4 specie) ones by comparing SSE (60% of *H.*

*portulacoides* and 9% of *S. maritima*) and WSW (33% of *H. portulacoides* and 35% of *S. maritima*) sectors during daytime emersion (Mann-Whitney tests, $p < 0.0001$). To the contrary, in summer, no significant difference in NEE fluxes was recorded between these two sectors (Mann-Whitney test, $p = 0.06$; Fig. 7) and more generally, between the different wind sectors (Table 1 and Fig. 7). For all seasons, during night-time emersion, we recorded that Southern sectors (ESE, SSE and SSW) emitted higher atmospheric $CO_2$ than Northern sectors (NNE and ENE), especially in winter and fall (Table 1 and Fig. 7)."

**Section 4.1. Marsh $CO_2$ uptake and influence of management practices** (this paragraph was modified in the revised MS)

p26, L554-567: "Moreover, despite of a lower benthic metabolism (photosynthesis and respiration) of muds than evergreen plants (Fig. 7), the microphytobenthos which can developed on mudflats (27% of the footprint area) could also significantly contribute to marsh production during daytime emersion, as highlighted in our studied salt marsh where static chamber measurements performed in March 2023 at midday showed a $CO_2$ uptake to a non-vegetated mudflat (NEE mean of -2.92 µmol m$^{-2}$ s$^{-1}$; unpublished results) and confirmed in an estuarine wetland in China (Xi et al., 2019). On an intertidal flat (France), EC measurements even showed a higher daily benthic metabolism with microphytobenthos (1.72 g C m$^{-2}$ d$^{-1}$; September/October 2007) than with *Zostera noltii* (1.25 g C m$^{-2}$ d$^{-1}$; July and September 2008), confirming the high biological productivity of mudflats (Polsenaere et al., 2012). Thus, the microphytobenthos could play a significant role in the atmospheric $CO_2$ uptake of salt marshes but also, more generally, in the carbon cycle of the coastal ocean because the resuspension of the microphytobenthos primary production during tidal immersion induce a large export of organic carbon from muds to coastal waters (up to 60% of the benthic primary production in a nearby tidal flat; Savelli et al., 2019). These fast-growing producers with high labile organic carbon could also be quickly degraded locally by microbial remineralization (Brouwer & Stal, 2001; Morelle et al., 2022) contrary to evergreen plants contributing to long-term "blue carbon" burial in sediments (Mcleod et al., 2011)."

**C4:** Are the results sufficient to support the interpretations and conclusions?

Mainly yes, although more than one year of data would be better.

We agree with Referee#2, more than one year of continuous EC data would be better. However, in this study, with solely the year 2020, we showed that (i) the temperate salt marsh was an annual carbon sink (-483 g C m$^{-2}$ yr$^{-1}$) mainly due to photosynthesis of halophile evergreen plants, (ii) light and temperature are the main controlling factors of NEE fluxes at long and short timescales, (iii) tidal marsh immersion reduced daytime $CO_2$ uptake and night-time $CO_2$ emissions and (iiii) the tidal rhythm did not affect the annual net C balance of the studied salt marsh. We felt that a year's worth of continuous data was sufficient to meet our initial objectives and add new information to the scientific literature. As mentioned in the submitted discussion, the other years of EC measurements (especially, year 2021) will be used to study specific processes and marsh $CO_2$ fluxes in relationship with the aquatic compartment (Mayen et al., in prep.) and the soil compartment (Arnaud et al., submitted 2022). Please refer to the submitted

MS, in the section 4.4. salt marsh carbon budgets for future research perspectives (p27-28, L628-646).

Would an area with more vascular plant cover give the same results?

The NEE analysis according to wind directions on daytime emersion periods showed that wind sectors with a higher vegetation density (South sectors) uptaked more atmospheric $CO_2$ than wind sectors with a high mudflat density (North sectors). Thus, based on this sectoral analysis of fluxes, a marsh area with more vascular plant cover could have a higher $CO_2$ sink capacity due to a higher benthic production rate than benthic microalgae on muds (microphytobenthos). However, due to the specific assemblage of our studied marsh (halophytic plants, mudflats and channels; Table 1), it remains complex to accurately study these habitat effects (vascular plants *vs.* microphytobenthos) on NEE fluxes at the marsh scale and draw more general conclusions. Ongoing atmospheric $CO_2$ exchange measurements are actually carried out since January 2023 up north over the Aiguillon intertidal Bay (France) where we precisely deployed an EC station at the edge between the tidal mud flat on the West side and salt marsh habitats on the East side of the footprint along with benthic chamber flux and water, sediment, soil carbon measurements and satellite analysis at each season to specially address these questions on relative habitat (mudflat *versus* salt marshes) influence on atmospheric $CO_2$ exchanges (Polsenaere, personal communication).

In light of sea level rise, marshes may turn into mudflats and have little primary production potential. Assessing areas with significant mud cover without comparing them to more densely vegetated areas may falsely highlight mudflats as a "desirable management practice."

We agree with Referee#2, it is important to assess the mudflat NEE fluxes on long-term series to better understand their capacity of net $CO_2$ uptake in intertidal systems such as salt marshes and tidal flats (Polsenaere et al., 2012; Savelli et al., 2019).

In our study, we showed a low benthic metabolism of mudflats during emersion with lower daytime $CO_2$ uptake than halophyle plants. With our EC approach, we did not directly show the role of mudflats in the metabolic fluxes of the marsh, since mudflats represents only 20% of the overall footprint and a maximum of 56% in the NNW sector. Moreover, microphytobenthos (fast-growing producers) is mostly constituted of labile organic carbon that could be quickly mineralised by heterotroph respiration locally whereas, evergreen pants (slow-growing producers) are mostly constituted of refractory organic carbon allowing significant long-term blue carbon burial. Thus, unlike microphytobenthos, a high density of vascular plants over salt marshes can be view of "desirable management practice" for carbon sequestration. In the revised manuscript, we discussed the mudflat's role in carbon fixation/emission and more generally, on the contribution of microphytobenthos on the coastal carbon cycle (see comment C3 responses above).

To better discuss the microphytobenthos *versus* salt marsh habitat influence of NEE fluxes, we will also add to our revised discussion first results concerning ongoing EC measurements currently carried out on a nearby intertidal bay (Aiguillon intertidal, France), at the edge

between the tidal mud flat (on the West side) and the salt marsh habitats (on the East side of the footprint) (see responses above).

**C5:** Is the description of experiments and calculations sufficiently complete and precise to allow their reproduction by fellow scientists (traceability of results)?

Yes, indubitably! The settings of the EC tower, instrumentation, gap-filling methods, preprocessing, and post-processing are described well.

We are grateful to Referee#2 for his positive feedback on the description of our EC measurements (section 2.2), EC data processing and quality control (section 2.4), and flux gap filling (2.5). Supplementary information on gap-filling method of EC data are done in the answers of Referee#1.

The methods sections could be shortened since the methods and techniques used are well-established in the literature. In many cases, citing the original authors of the methods and techniques should suffice.

In the revised manuscript, we reduced and summarised the section on the EC theory in the beginning of the section 2.2. Eddy Covariance and micrometeorological measurements (see also reviewer#1's responses). The EC methods and techniques are well-established in terrestrial ecosystem (Aubinet et al., 1999; Baldocchi, 2003; Aubinet et al., 2012; Burba, 2021) but not enough yet in intertidal ecosystems like tidal bays and salt marshes. Supplementary clarifications of the EC methodology were then added in the revised manuscript (footprint estimation, displacement height calculation, time average of turbulent EC data, etc.; see also reviewer#1's responses).

Assumptions regarding water elevation and time of inundation may be challenging to replicate the way it occurred in this study.

In the revised manuscript, we moved the explanation of water elevation and time inundation in the footprint analysis section (see 2.3. from L205 to L217). Due to our one-location water height measurements, we know that there is an incertitude related to the immersion of wind sectors within the footprint area but this important consideration was taken into account in our interpretations and further specified in the revised manuscript (see also reviewer#1's responses).

**C6:** Do the authors properly credit related work and indicate their new/original contribution?

Yes, the literature review is thorough; all the relevant and current papers are cited.

We thank Referee#2 for his positive feedbacks on the literature review and on the highlighting on the original contribution of our manuscript.

**C7:** Does the title clearly reflect the contents of the paper?

Yes, I have no problem with the title.

We thank Referee#2 for his positive feedbacks on the title of our manuscript.

**C8:** Does the abstract provide a concise and complete summary?

The Abstract is sufficient to raise curiosity about the study and reflects the content well.

We thank Referee#2 for his positive feedbacks on the abstract of our manuscript.

**C9:** Is the overall presentation well-structured and clear? Is the language fluent and precise?

The narrative is manageable with mathematical expressions and jargon. The authors take care to spell out the methods succinctly and efficiently.

We thank Referee#2 for his positive feedbacks on the overall presentation of our manuscript (structure, language, methods, etc).

The word immerged is used many times and needs to be checked. The correct word should be immersed.

In our revised manuscript, we replaced the word "immerged" by the word "immersed".

**C10:** Are mathematical formulae, symbols, abbreviations, and units correctly defined and used?

Yes, the representation is unambiguous and well-explained.

We thank Referee#2 for his positive feedbacks on the representation of mathematical formulae, symbols, abbreviations, and unit in our manuscript.

**C11:** Should any parts of the paper (text, formulae, figures, tables) be clarified, reduced, combined, or eliminated?

The colors of Figure 2 are hard to read in the legend. Patterns rather than colors better distinguish the plant species and cover types.

We have decided to retain the colours shown in Figure 2, as we believe they are sufficiently visible to be easily distinguished. However, we added additional pictures of different wind sectors in figure 2 to better visualize the plant species and cover types.

Figure 3 – The X-axis needs to be more legible

In the revised manuscript, x-axis of the figure 3 was modified to be more legible.

Figure 6 is the most crucial in the paper and needs to be better presented. It needs to be clearer and more readable. Clarify what the X values represent in this figure.

Referee#2 is right, figure 6 is important in our study to assess the influence of meteorological and hydrological drivers on NEE fluxes within five PAR groups. In the revised manuscript, x-axis of the figure 6 were modified and we have made significant efforts to better clarify this figure. Moreover, in this figure, we replaced RH by VPD (see Referee#1's response; Fig. 7-B).

The study area is diverse and mudflat covers a significant percent of the study area, especially in the ESE and WNW sections. Regarding the mudflat's role in carbon fixation/emission, it would be revealing how these areas compare to the more densely vegetated areas.

Referee#2 is right (see responses above), our studied salt marsh in terms of habitats in particular is diverse and complex, whose 27% of the studied footprint area was composed of mudflats. According to the land-use map, the highest mudflat covers are in the WNW (56%) and ENE (37%) wind sectors (Table 1; Fig. 2).

We agree with Referee#2, it is important to assess the metabolic fluxes of mudflats to better understand their $CO_2$ uptake capacity in intertidal systems such as salt marshes (see responses above). In the revised manuscript, the spatial NEE analysis according to wind directions during emersion periods showed that, generally, wind sectors with a high vegetation density (South sectors) uptaked and emitted more atmospheric $CO_2$ during daytime and night-time, respectively, than wind sectors with a high mudflat density (North sectors; Table 1 and Fig. 7). Thus, based on this sectoral analysis of NEE fluxes, mudflats areas on the Bossys perdus salt marsh could have a lower benthic metabolism in terms of production and respiration in comparison with more densely vegetated areas (Fig. 7). Moreover, in winter and fall at emersion, we highlighted significant higher $CO_2$ uptake rates of *H. portulacoides* than *S. maritima* whereas in summer, no significant difference was recorded between these two species. This seasonal difference could be related to the plant phenology with *H. portulacoides* as evergreen plant throughout the year whereas, the growing season for *S. maritima* was shorter

associated with a flowering period only from August to October. In the revised MS, we discussed more the mudflat's role in carbon fixation/emission in comparison with the marsh plant metabolism at emersion (see revised sections below).

**Section 3.4. Influence of environmental drivers on temporal NEE variations** (this paragraph was added in the revised MS)

p18, L435-447: "For wind directions, a spatial heterogeneity of NEE was recorded according to wind sectors both during daytime and night-time (Fig. 6-F). Within the footprint area composed of an assemblage of plants and muds (Fig. 2), the highest $CO_2$ uptakes were generally recorded from the Southern sectors (high vegetation:mud ratios) whereas, the lowest $CO_2$ uptakes were generally recorded from the Northern sectors (low vegetation:mud ratios; Fig. 7). For instance, our sectorial NEE analysis during daytime emersion showed that SSE sector (vegetation:mud ratio of 2.4; Table 1) uptaked 32% (winter), 25% (spring) and 50% (fall) times more atmospheric $CO_2$ than NNW sector (vegetation:mud ratio of 0.8; Table 1). Moreover, in winter and fall, we highlighted that $CO_2$ uptake rates of *H. portulacoides* (C3 specie) were significantly higher than *S. maritima* (C4 specie) ones by comparing SSE (60% of *H. portulacoides* and 9% of *S. maritima*) and WSW (33% of *H. portulacoides* and 35% of *S. maritima*) sectors during daytime emersion (Mann-Whitney tests, $p < 0.0001$). To the contrary, in summer, no significant difference in NEE fluxes was recorded between these two sectors (Mann-Whitney test, $p = 0.06$; Fig. 7) and more generally, between the different wind sectors (Table 1 and Fig. 7). For all seasons, during night-time emersion, we recorded that Southern sectors (ESE, SSE and SSW) emitted higher atmospheric $CO_2$ than Northern sectors (NNE and ENE), especially in winter and fall (Table 1 and Fig. 7)."

**Section 4.1. Marsh $CO_2$ uptake and influence of management practices** (this paragraph was modified in the revised MS)

p26, L554-567: "Moreover, despite of a lower benthic metabolism (photosynthesis and respiration) of muds than evergreen plants (Fig. 7), the microphytobenthos which can developed on mudflats (27% of the footprint area) could also significantly contribute to marsh production during daytime emersion, as highlighted in our studied salt marsh where static chamber measurements performed in March 2023 at midday showed a $CO_2$ uptake to a non-vegetated mudflat (NEE mean of -2.92 µmol m$^{-2}$ s$^{-1}$; unpublished results) and confirmed in an estuarine wetland in China (Xi et al., 2019). On an intertidal flat (France), EC measurements even showed a higher daily benthic metabolism with microphytobenthos (1.72 g C m$^{-2}$ d$^{-1}$; September/October 2007) than with *Zostera noltii* (1.25 g C m$^{-2}$ d$^{-1}$; July and September 2008), confirming the high biological productivity of mudflats (Polsenaere et al., 2012). Thus, the microphytobenthos could play a significant role in the atmospheric $CO_2$ uptake of salt marshes but also, more generally, in the carbon cycle of the coastal ocean because the resuspension of the microphytobenthos primary production during tidal immersion induce a large export of organic carbon from muds to coastal waters (up to 60% of the benthic primary production in a nearby tidal flat; Savelli et al., 2019). These fast-growing producers with high labile organic carbon could also be quickly degraded locally by microbial remineralization (Brouwer & Stal, 2001; Morelle et al., 2022) contrary to evergreen plants contributing to long-term "blue carbon" burial in sediments (Mcleod et al., 2011)."

**Section 4.2. Metabolism processes and controlling factors at multiple timescales** (this paragraph was modified in the revised MS)

p26-27, L574-581: "In our studied marsh, the halophyle vegetation, mostly composed of evergreen species, was autotrophic throughout the year allowing a net C uptake from the atmosphere during both the growing and non-growing seasons (between 9 g C m$^{-2}$ in December and 73 g C m$^{-2}$ in July) whereas, the senescence of smooth cordgrass plants in some salt marshes (*S. alterniflora* and *S. cynosuroides* for instance) from October produced a marsh heterotrophy and a net C source to the atmosphere in winter and fall (Schafer et al., 2014; Artigas et al. 2015; Forbrich et al., 2018). In our case, *S. maritima* is a perennial specie with a relatively short growing period, thus during winter and fall, the benthic metabolism of this halophytic plant could have a significant lower influence on marsh C uptake than *H. portulacoides* and *S. vera*."

I would welcome an assessment of the different cover types in the discussion. As Figure 2 and Table 1 show, wind direction is a sound basis for differentiating between the different mixture ratios of mud: water: vegetation – Figure 6E should be discussed in light of the numbers represented in Table 1.

In the revised manuscript, accordingly, we now present a sectorial analysis of NEE fluxes during daytime and night-time emersions (see figure 7 in the revised MS) to compare metabolic fluxes between high density plant areas (high plant:mud ratio) and high-density mudflat areas (low plant:mud ratio). Please refer to responses below (see sections 3.4. and 4.1. in the revised MS).

Also, it would be exciting to look at the plants in terms of C3, C4, or CAM and see if this affects their respective carbon assimilation rates.

In the revised MS, we added the metabolic pathways of our plants in term of C3 and C4 and discussed if this affect their carbon assimilation rates (see revised sections below).

**Section 2.1. Study site** (this paragraph was modified in the revised MS)

p6, L134-139: "The marsh vegetation assemblage was mainly composed by three halophytic species as perennial plants (*Halimione portulacoides*, *Spartina maritima* and *Suaeda vera*; Fig. 2) that associated with different metabolic pathways (the C3-type photosynthesis for *H. portulacoides* and *S. vera* and the C4-type photosynthesis for *S. maritima*; Duarte et al., 2013, 2014). Whereas *H. portulacoides* and *S. vera* are evergreen plants throughout the year, the growing season for *S. maritima* was shorter (from spring) with a flowering period between August and October (plants persist only in the form of rhizomes in winter and fall; Gernigon, personal communication)."

**Section 3.4. Influence of environmental drivers on temporal NEE variations** (these sentences were added in the revised MS)

p18, L441-447: "Moreover, in winter and fall, we highlighted that $CO_2$ uptake rates of *H. portulacoides* (C3 specie) were significantly higher than *S. maritima* (C4 specie) ones by comparing SSE (60% of *H. portulacoides* and 9% of *S. maritima*) and WSW (33% of *H. portulacoides* and 35% of *S. maritima*) sectors during daytime emersion (Mann-Whitney tests, $p < 0.0001$). To the contrary, in summer, no significant difference in NEE fluxes was recorded between these two sectors (Mann-Whitney test, $p = 0.06$; Fig. 7) and more generally, between the different wind sectors (Table 1 and Fig. 7). For all seasons, during night-time emersion, we recorded that Southern sectors (ESE, SSE and SSW) emitted higher atmospheric $CO_2$ than Northern sectors (NNE and ENE), especially in winter and fall (Table 1 and Fig. 7)."

**Section 4.1. Marsh $CO_2$ uptake and influence of management practices** (this paragraph was added in the revised MS)

p26, L547-553: "In salt marshes *H. portulacoides* (C3 specie) have high ability to acclimation to temperature variations and elevated $CO_2$, contrarily to *S. maritima* (C4 specie; Sousa et al., 2010). Indeed, increasing atmospheric $CO_2$ concentrations (from 380 to 760 ppm) produced an improvement of the light harvesting mechanisms and photosynthetic efficiency for C3 species whereas, negative impacts on photosynthetic ability of C4 species were recorded through photochemical and oxidative stress (Duarte et al., 2014). Thus, under future environmental conditions, the continuous atmospheric $CO_2$ increases due to human activities will favour the development of C3 species to the detriment of C4 species."

**Section 4.2. Metabolism processes and controlling factors at multiple timescales** (this paragraph was modified in the revised MS)

p26-27, L574-581: "In our studied marsh, the halophyle vegetation, mostly composed of evergreen species, was autotrophic throughout the year allowing a net C uptake from the atmosphere during both the growing and non-growing seasons (between 9 g C m$^{-2}$ in December and 73 g C m$^{-2}$ in July) whereas, the senescence of smooth cordgrass plants in some salt marshes (*S. alterniflora* and *S. cynosuroides* for instance) from October produced a marsh heterotrophy and a net C source to the atmosphere in winter and fall (Schafer et al., 2014; Artigas et al. 2015; Forbrich et al., 2018). In our case, *S. maritima* is a perennial specie with a relatively short growing period, thus during winter and fall, the benthic metabolism of this halophytic plant could have a significant lower influence on marsh C uptake than *H. portulacoides* and *S. vera*."

I would also welcome more discussion regarding the fact that the marsh – according to the data – remains a carbon sink over the winter month and what the expectations would be when the marsh is fully restored to its natural state.

The Bossys perdus salt marsh is not under on-going restoration. For several centuries, it was used for salt farming and oyster farming but since 1981, the salt marsh is protected within the

maritime part of the national natural reserve (NNR) with natural site hydrodynamics and marsh halophile vegetation preservation without major restoration work here (see Referee#1 comment responses). We modified our revised manuscript (see the revised section below) to give readers a better understanding of the history of the site and its current management practice.

**Section 2.1. Study site** (this paragraph was modified in the revised MS)

p4, L109-115: "The study was conducted at the Bossys perdus salt marsh situated along the French Atlantic coast on Ré Island (Fig. 1). It corresponds to a vegetated intertidal area of 52.5 ha that has been protected inside the National Natural Reserve (NNR) (Fig. 1). Between the 17th and 20th centuries, the salt marsh has experienced successive periods of intensive land-use (salt harvesting, oyster farming) and returns to natural conditions before becoming a permanent part of the NNR since 1981 for the biodiversity protection without major marsh restoration work (Gernigon, personal communication). It is currently managed to restore its natural hydrodynamics while conserving the site's specific typology due to past human activities (channel network, humps and dykes; Fig. 2)."

As suggested by Referee#2's comment, we further discussed the capacity of our halophytic vegetation to keep an atmospheric carbon sink over the winter month. Thus, in the revised manuscript, we discussed that our evergreen vegetation in the Bossys perdus salt marsh was mainly autotrophic throughout the year allowing a net $CO_2$ uptake during both the growing and non-growing seasons whereas the smooth cordgrasses as *S. alterniflora* in some tidal salt marshes was heterotrophic during winter and fall due to the plant senescence, producing, in turn, atmospheric $CO_2$ emissions during these periods of the year. However, the $CO_2$ sink in winter was mainly provided by the benthic metabolism of *H. portulacoides* rather than one of *S. maritima* due to its low growth activity during this period. Please refer to responses and revised sections above

**C12:** Are the number and quality of references appropriate?

Yes, it is well-balanced.

We thank Referee#2 for his positive feedbacks on the literature review in our manuscript.

**C13:** Is the amount and quality of supplementary material appropriate?

Yes, it is adequate for the study.

We are grateful to Referee#2 for his positive feedbacks on the supplementary materials of our manuscript.

---

## Author Response (AR2)

This is the second time I review the paper. The authors put a lot of effort in addressing my own and the second reviewer's comments. However, I have minor/editorial comments and two remaining scientific comments.

We are grateful to Referee#1 for his/her positive feedbacks on our revised manuscript. Comments from referees were very useful to improve the MS.

**Editorial:**
'specie' - the authors use 'C3/C4 specie', which I believe should be 'C3/C4 species'. Can be solved with a simple 'search and replace'

We have replaced "C3/C4 specie" by "C3/C4 species".

l. 593: 'did uptake' - can be replaced by 'took up' or similar

We have replaced "did uptake" by "took up" as proposed by Referee#1.

**Scientific:**
l. 283: I did not realize that in the original manuscript, but it looks to me that the authors did not apply a 'ustar filter' - otherwise there would be more nighttime data missing than 20%. This is routinely done in terrestrial system, with the goal to remove night time data collected under not well mixed conditions (because the flux measurement is not complete, and the ecosystem respiration estimate will be affected). It could be that at the coast this is not a big problem, but because it is routinely done, adding a sentence of explanation will help with comparison/repetition.

Referee#1 is right, we did not apply a ustar filter in our data processing as generally done in terrestrial ecosystem studies (Gu et al., 2005). Indeed, we measured only 11% of night-time EC data corresponding to a ustar threshold below 0.1 m s$^{-1}$ (mean wind speed of $1.15 \pm 0.52$ m s$^{-1}$) and above which NEE does not increase anymore with ustar values. This threshold value is lower than ranges determined in forests (0.2-0.4 m s$^{-1}$) and logically closer to values found in grassland (Gu et al., 2005). Contrary to terrestrial ecosystems (forests or agricultural cover), the low canopy height of the studied salt marsh (*Spartina maritima*, *Halimione portulacoides* and *Suaeda vera*) over the year strongly limits the $CO_2$ storage in the vegetation and, on the contrary, favours the atmospheric $CO_2$ circulation. Thus, with this 0.1 m s$^{-1}$ threshold, filtered night-time NEE data would be low; furthermore, this filter does not seem to affect monthly NEE versus Ta regressions presented in the MS endorsing our choice to do not apply a ustar filter on our measured EC data.

We have completed the revised MS for more precision concerning the ustar filter (see L244-L248, p 9).

Gu, L., Falge, E. M., Boden, T., Baldocchi, D. D., Black, T. A., Saleska, S. R., Suni, T., Verma, S. B., Vesala, T., Wofsy, S. C., and Xu, L.: Objective threshold determination for nighttime eddy flux filtering, Agricultural and Forest Meteorology, 128, 179–197, https://doi.org/10.1016/j.agrformet.2004.11.006, 2005.

Van Dam, B. R., Lopes, C. C., Polsenaere, P., Price, R. M., Rutgersson, A., and Fourqurean, J. W.: Water temperature control on $CO_2$ flux and evaporation over a subtropical seagrass meadow revealed by atmospheric eddy covariance, Limnol Oceanogr, 66, 510–527, https://doi.org/10.1002/lno.11620, 2021.

l. 602-607: I understand that the authors added this section in response to a comment by reviewer 2, but I think that it is not a relevant point to make for the system at hand. That is because the mix oc C3 and C4 plants at the study site follow a clear zonation, which seems to show that the C4 plant occurs at lower elevation (with more flooding of salt water), while the C3 plants occur at slightly higher elevation. This is rather typical for coastal wetlands, and C4 is associated with more salt tolerance (e.g. Bromham, L. and Bennett, T.H. (2014), Salt tolerance evolves more frequently in C4 grass lineages. J. Evol. Biol., 27: 653-659. https://doi.org/10.1111/jeb.12320). Thus, for a future marsh, the inundation regime is at least as important as atmospheric $CO_2$ levels. I would personally take out this section, because I don't think it is needed for the points the authors want to make, but I will leave it to them to expand on it.

Indeed, in response to a comment by Referee#2, we added in the revised MS the metabolic pathways of our plants in term of C3 and C4 and discussed if this affects their carbon assimilation rates. However, we agree with Referee#1 that this is not a major objective of our study. Consequently, we have removed this section that also lighten the discussion section.